# Genome-wide association meta-analyses and fine-mapping elucidate pathways influencing albuminuria

Alexander Teumer ⬤ et al.#

Increased levels of the urinary albumin-to-creatinine ratio (UACR) are associated with higher risk of kidney disease progression and cardiovascular events, but underlying mechanisms are incompletely understood. Here, we conduct trans-ethnic ($n = 564,257$) and European-ancestry specific meta-analyses of genome-wide association studies of UACR, including ancestry- and diabetes-specific analyses, and identify 68 UACR-associated loci. Genetic correlation analyses and risk score associations in an independent electronic medical records database ($n = 192,868$) reveal connections with proteinuria, hyperlipidemia, gout, and hypertension. Fine-mapping and trans-Omics analyses with gene expression in 47 tissues and plasma protein levels implicate genes potentially operating through differential expression in kidney (including *TGFB1*, *MUC1*, *PRKCI*, and *OAF*), and allow coupling of UACR associations to altered plasma OAF concentrations. Knockdown of *OAF* and *PRKCI* orthologs in *Drosophila* nephrocytes reduces albumin endocytosis. Silencing fly PRKCI further impairs slit diaphragm formation. These results generate a priority list of genes and pathways for translational research to reduce albuminuria.

#A full list of authors and their affiliations appears at the end of the paper.

Higher levels of the urinary albumin-to-creatinine ratio (UACR) are associated with adverse clinical outcomes, such as end-stage kidney disease, cardiovascular disease (CVD), and mortality[1–5]. Elevated UACR is a measure of kidney damage that is used to diagnose and stage chronic kidney disease (CKD)[6], which affects >10% of adults worldwide[7], and represents a hallmark of diabetic kidney disease[8]. Even moderate elevations in UACR predict poorer health outcomes, independently of the glomerular filtration rate[4,5]. Lowering of UACR by pharmacological inhibition of the renin–angiotensin–aldosterone system (RAAS) is considered renoprotective standard of care to slow CKD progression.[9–11] RAAS blockage is associated with a reduction of albuminuria and lower risk of end-stage kidney disease[12] and CVD events[10,13–15]. However, the risk of CVD events among CKD patients remains high[3]. A better understanding of the pathways related to the development and consequences of albuminuria may facilitate the search for novel therapies to treat or prevent CKD progression and CVD.

Levels of UACR have a heritable component in population-based studies and groups at high risk of CKD, such as certain indigenous populations or persons with diabetes[16–20]. However, the identification of genetic loci for UACR through genome-wide association studies (GWAS) has proven difficult, and detected loci showed variable effects across ancestries or disease groups[21]. Initial GWAS of UACR identified only two genome-wide significant loci, CUBN[22,23] and HBB[24]. A complementary approach using admixture mapping also identified the BCL2L11 locus[25]. One additional finding in patients with type I diabetes[26] was not detected in type II diabetes patients or the general population. Only very recently, a Mendelian Randomization study assessing a potentially causal effect of UACR on cardiometabolic traits based on data from the UK Biobank (UKBB) reported 33 genome-wide significant single-nucleotide polymorphisms (SNPs) associated with UACR[27]. The study supported a causal effect of higher UACR on elevated blood pressure and postulated that inhibition of UACR-increasing pathways could have anti-hypertensive effects and thereby reduce CVD risk.

In this project, we characterize known and identify additional novel genetic loci for UACR through trans-ethnic meta-analysis of GWAS from 564,257 participants, including an internal validation step and secondary analyses among participants with diabetes. To prioritize the most likely causal variants, genes, tissues, and pathways in associated loci, we perform functional enrichment analyses, statistical fine-mapping and integrative trans-Omics analyses, including with gene expression in 47 human tissues and plasma protein levels. Clinical correlates are identified through genome-wide genetic correlation analyses and a phenome-wide association scan of a genetic risk score for UACR in a large independent population. We evaluate translation to mechanistic insights in proof-of-concept studies for OAF and PRKCI using an experimental model of albuminuria. Together, the implicated variants, genes, proteins, tissues, and pathways provide a rich resource of new targets for translational research.

## Results
The workflow of our study, which identified 68 UACR-associated loci across primary and secondary analyses, is illustrated in Supplementary Fig. 1.

### Primary analysis: identification of 59 loci for UACR. The data based on 564,257 individuals from 54 studies were combined in a trans-ethnic meta-analysis of UACR, including 547,361 of European ancestry (EA), 6795 African Americans (AA), 6324 of East Asian ancestry, 2335 of South Asian ancestry, and 1442 Hispanics (Supplementary Data 1). The median of the median UACR across studies was 7.5 mg/g, and an average of 14.9% (range 3.2–70.9%) of participants had microalbuminuria (MA, UACR > 30 mg/g). Study-specific GWAS of UACR were carried out using imputed genotypes (Methods, Supplementary Data 2). We performed study-specific variant filtering and quality control (QC), followed by fixed-effects inverse-variance weighted meta-analysis. There was no evidence of unaccounted stratification (LD score regression intercept 0.95; genomic control (GC) parameter $\lambda_{GC}$ 1.03). Downstream analyses were based on 8,034,757 SNPs available after variant filtering (Methods). Using SNPs of minor allele frequency (MAF) > 1% across the genome, the heritability of UACR was estimated as 4.3%.

We identified 59 UACR-associated loci, defined as 1 Mb genomic segments carrying at least one SNP associated with UACR with $p < 5 \times 10^{-8}$ (Methods; Fig. 1, Supplementary Data 3). The index SNP mapped within 500 kb of previously reported index SNPs for UACR at 27 loci, considered known, and the remaining 32 loci were considered novel. These 59 SNPs explained 0.69% of the variance of the inverse normal transformed UACR residuals. There was little evidence of between-study heterogeneity (median $I^2$ statistic 3.2%; Supplementary Data 3), with all index SNPs showing an $I^2$ of <50%. In meta-regression analysis (Methods), none of the 59 index SNPs showed evidence of ancestry-related heterogeneity after multiple testing correction ($p < 8.5 \times 10^{-4}$, Fig. 1; Supplementary Data 3)[28]. Regional association plots of all loci are displayed in Supplementary Fig. 2.

Some of the loci contain biologically plausible candidates in addition to the known CUBN (cubilin) locus: for example, rare mutations in COL4A4 (Collagen Type IV Alpha 4 Chain) cause Alport syndrome, a monogenic disease of basement membranes that frequently leads to end-stage kidney disease. Recent sequencing studies show that the phenotypic spectrum of rare COL4A4 mutations extends to focal segmental glomerulosclerosis, which typically presents with proteinuria[29,30]. Our study extends the genetic spectrum to common COL4A4 variants associated with UACR in mostly population-based studies. Another example is NR3C2 (Nuclear Receptor Subfamily 3 Group C Member 2), which encodes the mineralocorticoid receptor that mediates aldosterone action. Pharmacological inhibition of the RAAS is the mainstay treatment to lower albuminuria, illustrating the potential for pharmacological intervention on pathways identified in this project.

Lastly, we estimated the number of expected discoveries and the corresponding percentage of GWAS heritability explained in future studies of yet larger sample size (Methods)[31] and found that such studies can be expected to detect additional UACR loci (Supplementary Fig. 3).

### Concordance between CKDGen cohorts and UK Biobank. To assess the influence of the UKBB, the largest study in the discovery sample ($n = 436,392$), we compared association statistics for the 59 index SNPs from the UKBB to the corresponding estimates from the 53 other studies participating in the CKDGen Consortium ($n \leq 127,865$). Effect direction was consistent for all 59 index SNPs ($p_{binomial\ test} = 3.5 \times 10^{-18}$; Fig. 2a), and 53 showed nominally significant associations in the CKDGen cohorts alone ($p < 0.05$; Supplementary Data 4). Two loci with strong effects in UKBB but not significant in CKDGen were AHR (aryl hydrocarbon receptor) and CYP1A1 (Cytochrome P450 Family 1 Subfamily A Member 1), potentially reflecting factors related to standardized sample handling, storage, and measurements in the UKBB, or population-specific exposures.

### Secondary ancestry-specific and diabetes-specific analyses. First, we conducted ancestry-specific meta-analyses for EA ($n =$

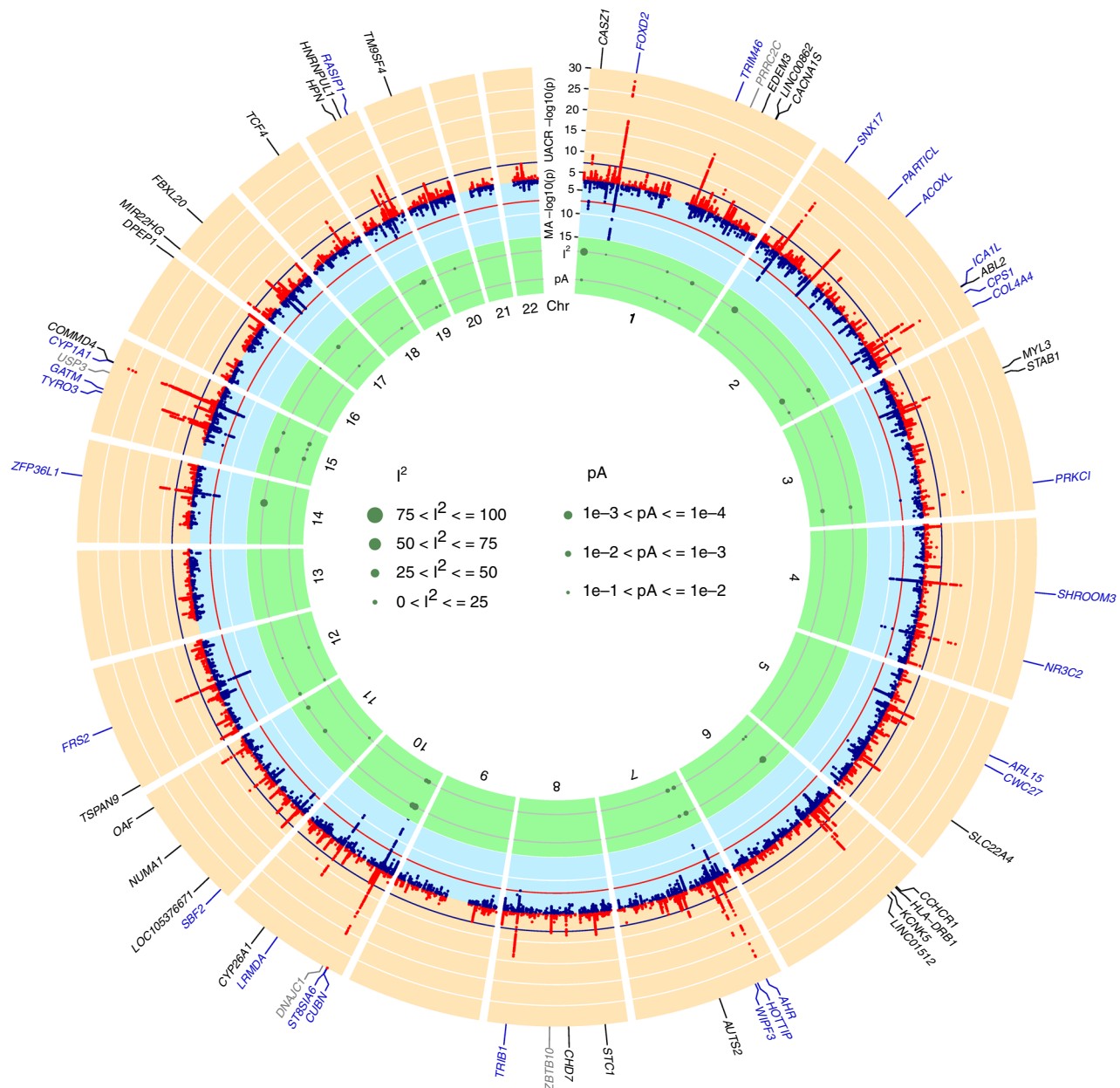

**Fig. 1** Genome-wide association results. The circos plot provides an overview of the association results: Red band: $-\log_{10}(p)$ for association in the trans-ethnic meta-analysis of urinary albumin-to-creatinine ratio (UACR), ordered by chromosomal position. The blue line indicates genome-wide significance ($p = 5 \times 10^{-8}$). Black gene labels indicate novel loci, blue labels indicate known loci (known index SNP within ± 500 kb region of current index SNP), gray labels indicate loci not associated with UACR at the nominal significance level ($p \geq 0.05$) in the 53 CKDGen cohorts without UKBB. Blue band: $-\log_{10}(p)$ for association with microalbuminuria (MA), ordered by chromosomal position. The red line indicates genome-wide significance ($p = 5 \times 10^{-8}$). Green band: measures of heterogeneity related to the UACR-associated index SNPs, where the dot sizes are proportional to two measures of heterogeneity, $I^2$ and the $-\log_{10}(p)$ for heterogeneity attributed to ancestry (pA)

547,361) and for AA ($n = 6795$), where ancestry-specific loci have been described[32,33]. There was little evidence of inflation of the results ($\lambda_{GC}$ 1.06 for AA and 1.01 for EA; Methods). These meta-analyses identified 61 loci in EA, of which 56 overlapped with those from the primary trans-ethnic meta-analysis (Supplementary Data 5 and further discussed below), and no genome-wide significant loci in AA. The known UACR-associated sickle cell trait variant rs334 in *HBB* showed suggestive association in the AA-specific analysis ($p = 6.1 \times 10^{-8}$).

The other secondary analysis was restricted to 51,541 individuals with diabetes, in whom a larger effect of the known *CUBN* locus has been reported[23]. This analysis identified eight

loci (Supplementary Fig. 4), four of which were not detected in the primary meta-analysis (*KAZN* [Kazrin, Periplakin Interacting Protein], *MIR4432HG-BCL11A*, *FOXP2*, and *CDH2*). Internal validation of the UKBB ($n = 21,703$) and CKDGen cohorts ($n \leq 29,812$) statistics found the effects to be direction consistent, of similar magnitude and at least nominally significant in both subsets at all eight loci (Supplementary Data 6). Index SNPs at *CUBN* and *HPN* (Hepsin) showed larger effect sizes among those with diabetes compared with the overall sample (Supplementary Data 6). Among the novel loci, it is noteworthy that *BCL11A*, a transcriptional regulator of insulin secretion[34], is involved in fetal-to-adult globin switching, as is the known UACR risk gene

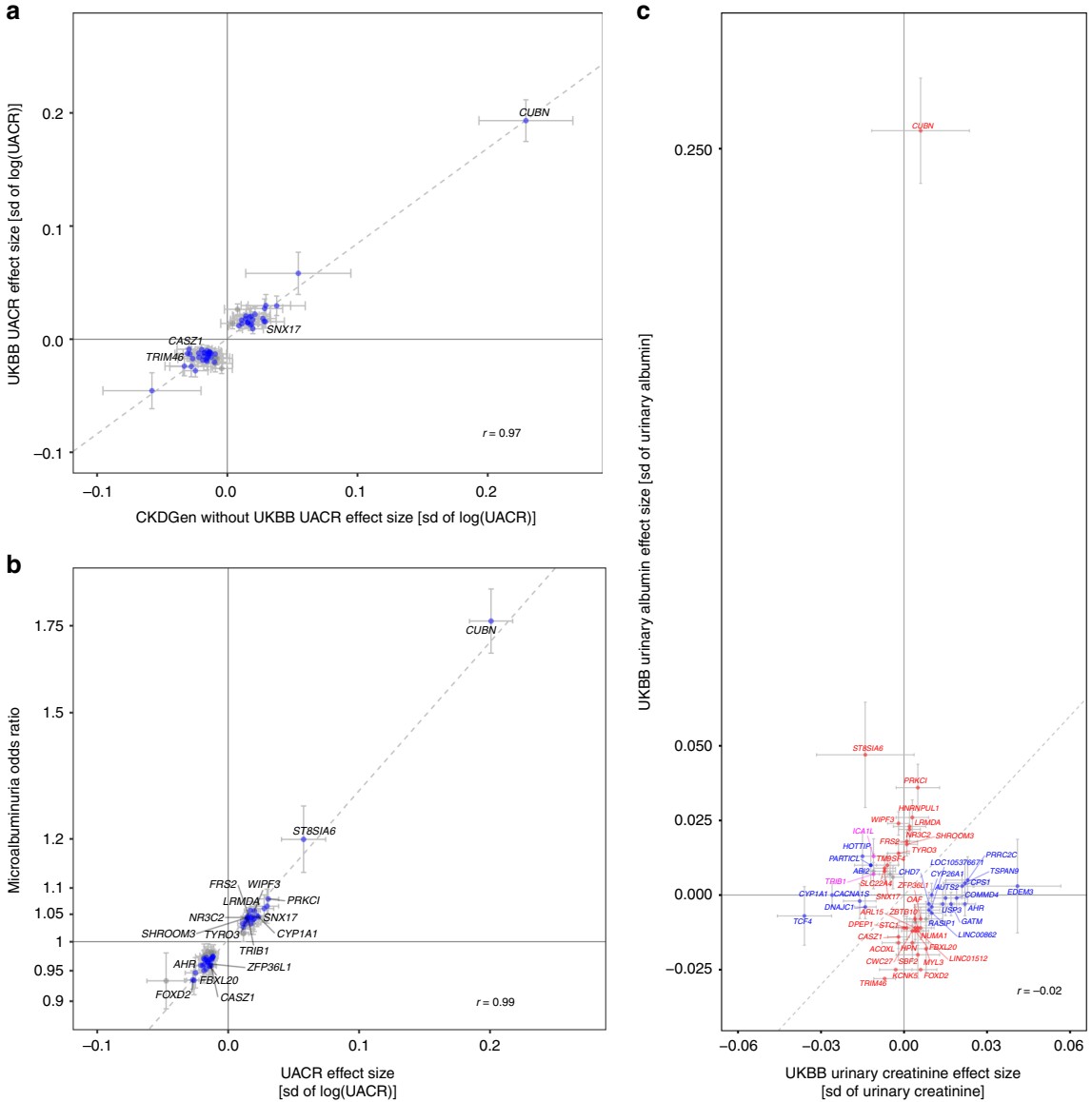

**Fig. 2** Internal concordance of the urinary albumin-to-creatinine ratio (UACR) results, and association with microalbuminuria, urinary creatinine and albumin. **a** Comparison of effect estimates of the 59 genome-wide significant trans-ethnic UACR index SNPs in the UKBB (x-axis) and in the CKDGen cohorts without UKBB (y-axis). Blue dots indicate nominal significance ($p < 0.05$) in the CKDGen cohorts without UKBB, and loci at genome-wide significance ($p < 5 \times 10^{-8}$) in that meta-analysis are labeled with the closest gene. **b** Comparison of effect estimates of the 59 trans-ethnic UACR index SNPs (x-axis) with their corresponding estimate from the GWAS of microalbuminuria (MA; y-axis). Blue dots indicate significance in the MA results after multiple testing correction ($p < 0.05/59 = 8.5 \times 10^{-4}$), and loci that achieved genome-wide significance ($p < 5 \times 10^{-8}$) for MA are labeled. In both panels, the dashed line represents the line of best fit through the effect estimates. **c** Comparison of effect estimates of the 59 genome-wide significant trans-ethnic UACR index SNPs for their effect on urinary creatinine (x-axis) and urinary albumin levels (y-axis) in the UKBB sample. Blue, red, and purple color indicate significant associations after multiple testing correction ($p < 0.05/59 = 8.5 \times 10^{-4}$) with urinary creatinine, urinary albumin, and both, respectively. Significant associations are labeled with the closest gene name. The dashed line represents the median $y = x$. In all panels, error bars indicate 95% confidence intervals (CIs), and the Pearson correlation coefficient $r$ between the effect estimates is shown. The effect directions correspond to the effect allele of the trans-ethnic UACR meta-analysis results

*HBB. KAZN* encodes for a protein with a role in actin organization and adhesion[35] that is highly abundant in glomeruli. QQ plots and Manhattan plots of the secondary meta-analyses are shown in Supplementary Figs. 5 and 6.

**Functional enrichment and pathways.** We searched for tissues, cell types, and systems that are enriched for the expression of genes mapping to the UACR-associated loci (Methods)[36]. Based on all SNPs with $p < 5 \times 10^{-8}$ from the trans-ethnic meta-

analysis, there was no significant (false discovery rate [FDR] < 0.05) enrichment after correction for multiple testing (Supplementary Data 7). Nominally significant associations ($p < 0.05$) were observed for 37 annotations mapping into six systems (urogenital including kidney, endocrine, digestive including liver, musculoskeletal, respiratory, sense organs; Supplementary Fig. 7) and five tissues (exocrine glands, prostate, mucous membrane, membranes, and respiratory mucosa). These results reveal plausible enrichments although they did not reach significance after correction for multiple testing.

Next, we evaluated whether reconstituted gene sets were significantly (FDR < 0.05) enriched for genes mapping to UACR-associated loci, and identified three sets with FDR < 0.01 (embryonic development, partial embryonic lethality during organogenesis, abnormal placental labyrinth vasculature morphology). The remaining significant gene sets included terms that can be reconciled with existing knowledge about albuminuria, including "tube development", "abnormal kidney morphology", and several terms related to vascular development and morphology (Supplementary Data 8).

**UACR-associated loci are associated with MA.** Clinical MA (UACR > 30 mg/g) is associated with increased risk for adverse kidney and cardiovascular outcomes, as well as mortality[3]. We therefore evaluated the association of the 59 UACR index SNPs with MA by meta-analyzing data from 36 cohorts and 347,283 individuals (Supplementary Data 1; Fig. 1). Figure 2b shows that for all UACR index SNPs, the allele associated with higher UACR was associated with an increased risk of MA (Supplementary Data 3). Of the 59 SNPs, 49 were significantly associated with MA after correction for multiple testing ($p < 0.05/59 = 8.5 \times 10^{-4}$), including 17 that reached genome-wide significance. The low-frequency missense SNP rs45551835 in *CUBN* showed the largest effect with an odds ratio (OR) of 1.76 (95% CI 1.67–1.87) per minor allele. When 232,751 UKBB participants were grouped into quartiles based on a UACR genetic risk constructed from the 59 index SNPs, each quartile showed a significantly higher OR for MA compared with the lowest quartile (e.g., OR of 1.69 for quartile 4 vs. 1, $p = 3.0 \times 10^{-191}$, Supplementary Table 1).

**UACR loci: association with urinary albumin and creatinine.** The UACR is a ratio. Understanding whether a genetic locus is more strongly associated with its numerator, albumin, or with its denominator, creatinine, may provide important physiological insights. We therefore performed separate tests for urinary albumin and creatinine in the UKBB sample ($n_{Ualbumin} = 436,398$; $n_{Ucreatinine} = 436,412$). Of the 59 index SNPs, 31 were significantly associated with urinary albumin ($p < 8.5 \times 10^{-4}$), 21 with urinary creatinine, and two with both. The *CUBN* locus showed the largest effect on urinary albumin, and was not significantly associated with urinary creatinine levels (Fig. 2c), followed by *ST8SIA6* (ST8 alpha-N-acetyl-neuraminide alpha-2,8-sialyltransferase 6), *PRKCI* (protein kinase C iota), *TRIM46/MUC1* (Mucin 1, cell surface associated), *HNRNPU L1/TGFB1* (transforming growth factor beta 1), *FOXD2*, *KCNK5*, *WIPF3* (WAS/WASL interacting protein family member 3), *LRMDA*, and *NR3C2*.

**A genetic UACR score is associated with medical diagnoses.** Next, we evaluated whether a weighted genetic risk score (GRS) composed of UACR-increasing alleles was associated with clinical endpoints in a large, independent electronic medical record database to detect diagnoses with potentially shared genetic components or co-regulation. We tested associations with 1422 billing code-based phenotypes of up to 192,868 EA participants of the Million Veteran Program (MVP) from US Veterans' Administration facilities[37]. Significant associations ($p < 3.5 \times 10^{-5}$, 0.05/1,422) were detected with 10 diagnoses: proteinuria, four related to hyperlipidemia, two related to hypertension, two related to gout, as well as Fuchs' dystrophy (Fig. 3). While the association with disorders of lipoid metabolism had the lowest *p*-value ($p = 4.1 \times 10^{-11}$), the association with Fuchs' dystrophy showed the greatest magnitude (OR = 6.68 per SD increase of log [UACR], 95% CI 3.06–14.59, $p = 1.9 \times 10^{-6}$), followed by proteinuria (OR = 2.7, 95% CI 1.76–4.14, $p = 5.0 \times 10^{-6}$). Many

other associations that approached statistical significance were related to the kidney and metabolic diseases (Supplementary Data 9).

The association with Fuchs' disease, a dystrophy of the corneal endothelium, was unexpected and assessed in greater detail. Autosomal-dominant forms of Fuchs' dystrophy have been attributed to genetic variation in *TCF4* (transcription factor 4)[38], a novel UACR-associated locus identified here (index rs11659764, $p = 2.8 \times 10^{-11}$; $r^2 = 0.21$, D' = −0.97 with rs613872, a previously reported Fuchs index SNP[39]). After exclusion of the *TCF4* index SNP, the GRS was still significantly associated with proteinuria, hyperlipidemia codes, gout, and hypertension with nearly identical ORs, but the association with Fuchs' dystrophy disappeared ($p = 0.2$). This illustrates that unexpected significant associations from PheWAS require careful evaluation.

We also evaluated an association of the GRS with cardiovascular outcomes based on published GWAS and the UKBB (Supplementary Table 2). This revealed significant ($p < 0.007$, Methods) positive associations of the GRS with an increased risk of hypertension ($p = 2.4 \times 10^{-21}$). Conversely, weighted genetic risk scores based on recently published GWAS of systolic and diastolic blood pressure as well as of type 2 diabetes were positively associated with UACR ($p = 3.5 \times 10^{-63}$ for systolic and $p = 1.2 \times 10^{-24}$ for diastolic blood pressure, $p = 1 \times 10^{-10}$ for type 2 diabetes; Supplementary Table 2).

**Genome-wide genetic correlations of UACR.** Albuminuria is associated with multiple cardiovascular and metabolic traits and diseases[4,40–42]. In addition to the GRS analyses, we thus also assessed genome-wide genetic correlations between the EA-specific UACR association statistics and 517 traits and diseases (Methods; Supplementary Data 10). Significant genetic correlations ($p < 9.7 \times 10^{-5}$ [0.05/517]) were observed for 67 traits (Fig. 4). The strongest negative correlations were observed for urinary creatinine and other urinary parameters, and the largest positive genetic correlations with different measures of hypertension. These findings provide support for the observational association between albuminuria and blood pressure on a genetic level, the significant associations between the UACR GRS and hypertension in the MVP population, and the recent Mendelian Randomization study of UACR[27]. Negative genetic correlations with anthropometric measures are potentially explained by their positive associations with muscle mass, and hence creatinine concentrations.

**Statistical fine-mapping and secondary signal analysis.** Statistical fine-mapping was performed using summary statistics to prioritize SNPs or sets of SNPs (credible set) driving each association signal (Methods). These analyses were limited to EA, comprising > 97% of the total sample, for whom large data sets to estimate reference LD for summary statistics-based fine-mapping were publicly accessible[43,44]. Based on 57 combined genomic regions from the 61 genome-wide significant loci in EA (Methods, Supplementary Data 5), we identified 63 independent SNPs (Supplementary Data 11). Next, 99% credible sets were computed based on Approximate Bayes Factors, resulting in a set of SNPs that with 99% posterior probability (PP) contained the variant(s) driving the association signal for each of the 63 conditionally independent signals[45]. The credible sets contained a median of 25 SNPs (Quartile 1: 10; Quartile 3: 74). Two credible sets at *CUBN* and one at *PRKCI* consisted of a single SNP (Supplementary Data 12). The previously described *CUBN* missense SNP rs45551835 (p.A2914V) had a PP of causing the association signal of >99.9%. There were 11 small credible sets with ≤5 SNPs, representing candidate causal variants for further study.

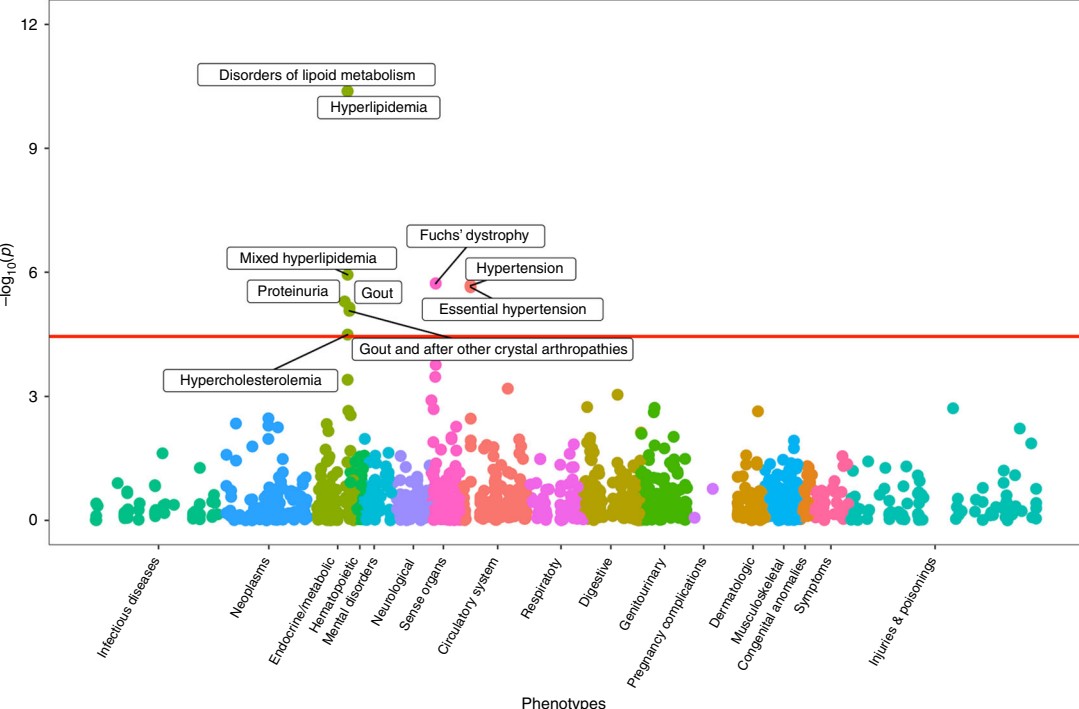

**Fig. 3** Phenome-wide association scan of a genetic urinary albumin-to-creatinine ratio (UACR) risk score. PheWAS association results were obtained from EA participants of the Million Veteran Program. Association test -$\log_{10}(p$-values) are plotted on the $y$-axis, and the corresponding trait or disease category on the $x$-axis. Significant results, after correcting for the 1422 phenotypes tested ($p < 0.05/1422 = 3.5 \times 10^{-5}$), are labeled in the figure

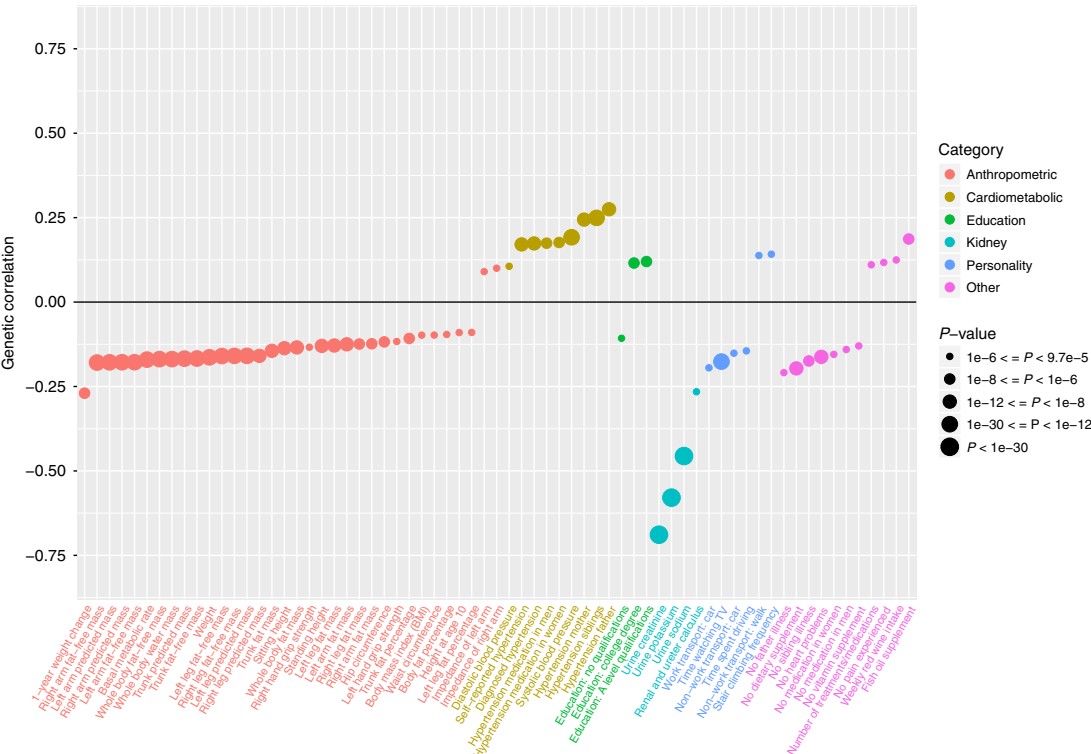

**Fig. 4** Genetic correlation of urinary albumin-to-creatinine ratio (UACR) with other traits and diseases. Significant ($p < 9.7 \times 10^{-5}$) genetic correlations based on the genome-wide summary statistics from the EA UACR GWAS and 517 pre-computed and publicly available GWAS summary statistics of UKBB traits and diseases, available through LDHub. Traits are shown on the $x$-axis, and colored according to broad physiological categories. Genetic correlations between traits and UACR are reported on the $y$-axis. Dot size is proportional to the –$\log_{10}(p)$ of the corresponding genetic correlation

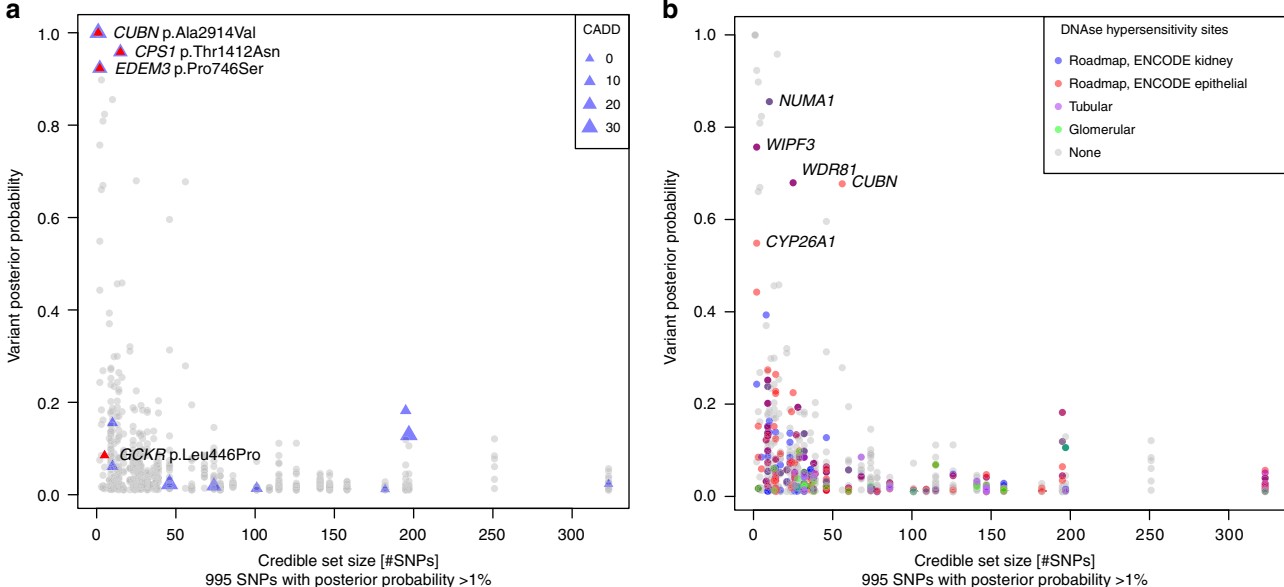

**Fig. 5** Fine-mapping and functional annotation of potentially causal variants. Overview of 995 SNPs with a posterior probability of association with urinary albumin-to-creatinine ratio (UACR) of >1%. The x-axis indicates the 99% credible set size and the y-axis the SNPs' posterior probability of association. In panel **a**, missense SNPs are marked by triangles, with size proportional to the SNP CADD score. In panel **b**, SNPs are color-coded with respect to location in regulatory regions of specific kidney tissues. The labels show the closest gene, and are restricted to variants mapping to small credible sets (≤5 SNPs), or to variants with high individual posterior probability (>0.5) of driving the association signal. For the *CUBN* locus, a credible set was computed for each independent SNP

All 995 SNPs with PP > 1% were annotated. Regulatory potential was assessed via mapping into regions of open chromatin identified from primary cultures of human tubular and glomerular cells (GEO accession number GSE115961)[46] and from publicly available kidney cells types (ENCODE and Roadmaps Projects; Methods). Supplementary Data 12 summarizes annotation information for all variants with PP > 1% that mapped into small credible sets or those containing a SNP with PP > 50%. Among these, there were four missense SNPs in *CUBN*, *CPS1*, *EDEM3*, and *GCKR* (Fig. 5a; Supplementary Table 3). One non-exonic SNP near *NUMA1* with PP > 50% mapped into open chromatin in both glomerular and tubular primary cell cultures, and four other SNPs in or near *WIPF3*, *WDR81*, *CUBN*, and *CYP26A1* mapped into putative regulatory regions in other kidney tissues or cell lines (Fig. 5b, Supplementary Data 12).

**Association with gene expression and co-localization**. We investigated whether the UACR-association signals co-localized with association signals for transcript abundance of any genes in *cis* across 47 tissues, thereby implicating effector genes at associated loci (Methods). Gene expression was quantified via RNA-seq in 44 tissues from the GTEx Project [https://gtexportal.org/] and in kidney cortex from The Cancer Genome Atlas[47], and via microarray from microdissected glomerular and tubulointerstitial portions of kidney biopsies from participants of the NEPTUNE study[48] (Methods).

We identified nine genes for which *cis* eQTLs in kidney tissues co-localized with the UACR association signals with a high PP (≥80%), implicating a shared underlying variant (Fig. 6). These represent candidate causal genes for further investigation (Table 1). Alleles associated with higher UACR were associated with higher expression of *MUC1* and *PRKCI* across a range of tissues. This observation is consistent with a gain-of-function mechanism proposed for the monogenic kidney disorder caused by *MUC1* variation[49]. Conversely, alleles associated with higher UACR were associated with lower *OAF* and *TGFB1* expression.

The co-localization with expression of *WIPF3* in glomerular kidney portions illustrates an example of a potentially regulatory causal variant, rs17158386, which maps into open chromatin in kidney tissue (Figs. 5b, 6). Across kidney tissues, co-localization was most often observed in glomerular kidney portions, consistent with the prominent role of the glomerular filtration barrier in albuminuria. Altogether, there were 90 significant co-localizations in at least one of the 47 evaluated tissues (Supplementary Fig. 8).

Association with gene expression in *trans* requires large sample sizes and was thus evaluated for all index SNPs in whole blood. Excluding the extended MHC region, there was one SNP associated with expression of one or more transcripts in *trans* in more than one study (Supplementary Table 4): genotype at rs12714144, upstream of *PARTICL* on chromosome 2, was associated with the expression of *DPEP3*, encoded on chromosome 16.

**Association with protein levels and co-localization analyses**. Recently, large GWAS of plasma protein levels have been published, which allow for systematic investigations of associated variants (pQTLs). Using these data, we investigated the association of the 61 EA index SNPs in a pQTL study of 3301 healthy EA participants of the INTERVAL study[50]. Genome-wide significant associations were identified between 17 UACR-associated SNPs and plasma levels of 53 unique proteins, for a total of 56 associations (Supplementary Data 13). Interestingly, concentrations of three proteins each showed associations with two UACR-associated index SNPs on different chromosomes, thereby connecting the two genetic loci through association with plasma concentrations of the same protein: SNPs rs34257409 on chromosome 1 and rs838142 on chromosome 19 with plasma gastrokine-2 (GKN2) concentrations, rs12714144 on chromosome 2 and rs1010553 on chromosome 3 with concentrations of Janus kinase and microtubule interacting protein 3 (JAKMIP3), and rs1010553 on chromosome 3 and rs2954021 on chromosome

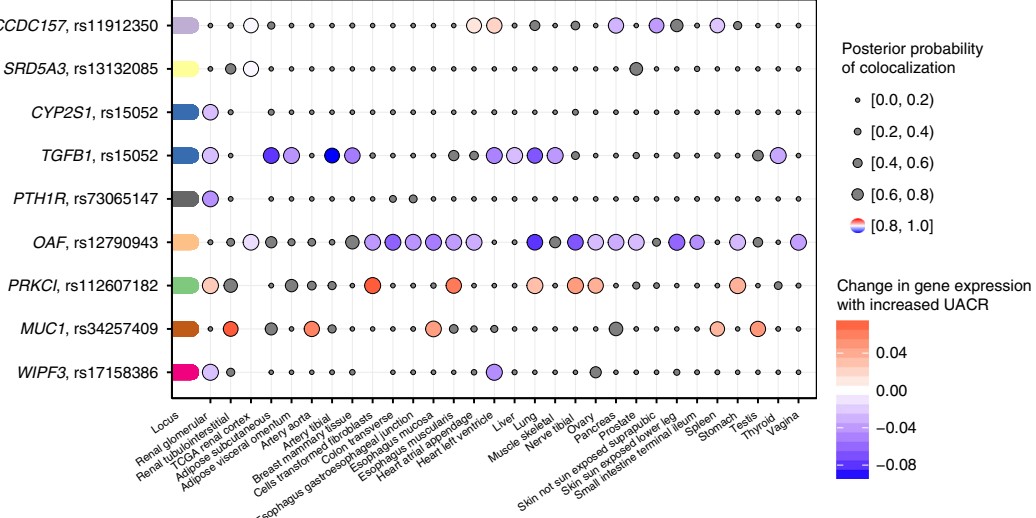

**Fig. 6** Co-localization of associations signals for urinary albumin-to-creatinine ratio (UACR) and gene expression in kidney tissues. The plot shows the nine genes for which there is a high likelihood (posterior probability ≥ 80%) of a shared causal signal for gene expression in at least one of three kidney tissues and UACR. The loci are colored-coded and shown on the y-axis with the closest gene next to the index SNP. Co-localization with gene expression across all tissues (x-axis) is shown as dots, where the size of the dots (implying that eQTL data were available) corresponds to the posterior probability of the co-localization. The change in UACR is color-coded relative to the change in gene expression, or gray in case of a posterior probability < 80%

8 with inter-alpha-trypsin inhibitor heavy chain 1 (ITIH1) concentrations.

Co-localization of UACR association signals with those for pQTLs of 38 proteins (Methods, Supplementary Table 5) provided evidence for a shared underlying SNP for plasma concentrations of the Out At First Homolog (OAF) protein. This was consistent with the eQTL co-localization analyses, with the minor T allele at rs12790943 associated with higher levels of UACR as well as with both lower *OAF* transcript levels in multiple tissues and lower OAF plasma levels (Fig. 7). Association patterns with UACR (Fig. 7a) and *OAF* transcript levels (Fig. 7b) looked similar, as expected for a shared underlying variant. The pattern looked different for OAF plasma levels, and conditional analyses revealed two independent SNPs (rs117554512 and rs508205; $r^2 = 0$, $D' = 0.02$ in the 1000 Genomes Project EUR sample). There was no evidence for a shared variant underlying the associations of UACR and OAF plasma levels for the signal tagged by the initial index SNP for OAF plasma levels, rs117554512 (PP H4 = 0; Fig. 7c), which was also significantly associated with plasma levels of IL25 in *trans* ($p = 1.3 \times 10^{-12}$, Supplementary Data 13). Conversely, there was strong evidence for a shared variant underlying associations with UACR and OAF plasma levels tagged by the second, independent signal at rs508205 (PP H4 = 0.99; Fig. 7d), allowing to follow associations from genetic variants to transcript, protein, and phentotype. The SNP rs508205 is located upstream of *OAF*, and was also the index variant identified in the trans-ethnic meta-analysis of UACR ($r^2 = 0.94$ with rs12790943 in the 1000 Genomes Project EUR sample). It represents an interesting regulatory candidate variant because of its relatively small credible set of eight SNPs, a CADD score of 13, and its localization in open chromatin in kidney tissue.

**In vivo analyses of Drosophila orthologs.** Finally, we used a *Drosophila* model to establish proof-of-principle that prioritized candidates can be used to gain mechanistic insights into albuminuria. *Drosophila* nephrocytes are specialized cells that harbor a slit diaphragm formed by the orthologs of the mammalian slit diaphragm proteins. These cells exhibit size-dependent molecule filtration across the slit diaphragm, followed by endocytosis via

the scavenger receptor Cubilin and finally lysosomal degradation or storage. Protein endocytosis mainly occurs within a network of membrane invaginations, the labyrinthine channels. Formation of the labyrinthine channels depends on presence of functional slit diaphragms. Thus, these cells reflect aspects of glomerular (slit diaphragm) and proximal tubular function (protein endocytosis)[51]. Studying endocytosis of a tracer molecule able to pass the slit diaphragm, such as albumin, renders an integrative read-out of nephrocyte function[52]: FITC-albumin uptake declines both through loss of slit diaphragms and also through impaired protein endocytosis. We selected three candidates for functional study, based on their associations with urinary albumin (Fig. 2c), support from downstream fine-mapping and co-localization analyses (Table 1), and degree of conservation and availability of at least two independent *Drosophila* RNAi lines per gene: *OAF*, *PRKCI*, and *WIPF3*. Orthologs of *OAF* (*oaf*), *PRKCI* (*aPKC*), and *WIPF3* (*Vrp1*) were silenced specifically in nephrocytes by crossing *Dorothy-GAL4* with the respective UAS-RNAi line.

Nephrocytes stained with an available antibody for aPKC showed a strongly reduced signal using two independent *aPKC*-RNAi lines (Supplementary Fig. 9A–C). We observed no effect of *Vrp1*-RNAi on nephrocyte function studying FITC-albumin endocytosis (Supplementary Fig. 9D, E). In contrast, we detected a significant reduction of tracer endocytosis upon silencing *oaf* and *aPKC* (Fig. 8a, b). This indicates a functional requirement of these genes within nephrocytes and supports a role of their human orthologs in glomerular filtration or tubular re-uptake of albumin. To distinguish between these roles, we studied immunofluorescence of the *Drosophila* slit diaphragm proteins, whose staining patterns remain unaltered in isolated defects of protein endocytosis. Despite the significant impairment of nephrocyte function, we observed a slit diaphragm staining pattern comparable to control conditions for *oaf*-RNAi (Fig. 8c–f). This suggests that oaf may be dispensable for slit diaphragm formation, but likely is involved in protein reabsorption. Accordingly, co-localization with *OAF* gene expression in human kidney was observed in the renal cortex, reflecting largely tubulointerstitial portions, and protein staining in the Human Protein Atlas is observed in tubules but not glomeruli. Conversely, silencing the ortholog of *PRKCI* entailed an extensive

**Table 1 Evidence for candidate causal genes at UACR-associated variants**

| Gene | SNP | H4 coloc | Credible set size | SNP PP | Functional consequence | CADD | DHS | Brief summary of literature and gene function |
|---|---|---|---|---|---|---|---|---|
| PRKCI | rs112607182 | 1.00 | 1 | 1.00 | Intergenic, downstream | 1.9 | – | PRKCI encodes a serine/threonine protein kinase that plays a role in microtubule dynamics. Has been identified as an important factor for actin cytoskeletal regulation in podocytes (PMID: 24096077). Podocyte-specific deletion of aPKClambda/iota in mice results in severe proteinuria (PMID: 19279126). |
| TGFB1 | rs15052 | 1.00 | 3 | 0.75 | 3'UTR (HNRNPUL1) | 9.9 | – | TGFB1 encodes a transcription factor that controls proliferation, differentiation and other functions in many cell types. Has been implicated as a cause of fibrosis in most forms of experimental and human kidney disease (PMID 10793168). Numerous publications and animal models connect it to diabetic kidney disease, as well as numerous animal models. 1*, 2*, 3* |
| WIPF3 | rs17158386 | 1.00 | 2 | 0.81 | Intergenic | 11.6 | – | The protein encoded by WIPF3 is involved in the Cdc42/N-WASP/Arp2/3 signaling pathway-mediated remodeling of the actin cytoskeleton (PMID: 11553796). |
| PTH1R | rs73065147 | 0.98 | 14 | 0.20 | Intergenic | 15.1 | – | PTH1R encodes for a receptor for parathyroid hormone, with high expression only in kidney cortex. The PTHrP/PTH1R system appears to adversely affect the outcome of diabetic and other renal diseases (PMID: 16783882, 21052497). Rare mutations have been reported to cause multiple aut-rec (#215045, #600002), or aut-dom (#125350, #156400) chondrodysplasias or tooth eruption phenotypes. |
| CYP2S1 | rs15052 | 0.95 | 3 | 0.75 | 3'UTR (HNRNPUL1) | 9.9 | – | CYP2S1 encodes for a member of the cytochrome P450 enzyme family, which catalyze many reactions involved in drug and lipid metabolism. It is transcriptionally regulated by AHR, also identified in the present GWAS meta-analysis, in rats (PMID: 19883719). |
| MUC1 | rs34257409 | 0.89 | 25 | 0.10 | Intergenic | 3.1 | 1* | MUC1 encodes for a membrane-bound member of the mucin family that play an essential role in forming protective mucous barriers on epithelial surfaces. Rare mutations cause medullary cystic kidney disease 1 (#174000), an autosomal-dominant tubulo-interstitial kidney disease. Patients show minimal to mild proteinuria in addition to decreased eGFR and renal cysts (PMID: 29217307). |
| OAF | rs12790943 | 0.97 | 7 | 0.47 | Intergenic | 1.8 | 1* | The OAF gene encodes for a transcription factor of the basic helix-loop-helix family. Relatively little is known about its function in humans. |
| SRD5A3 | rs13132085 | 0.92 | 183 | 0.03 | Intergenic | 4.0 | – | The protein encoded by SRD5A3 gene is involved in the production of androgen 5-alpha-dihydrotestosterone, and in the conversion of polyprenol into dolichol and thereby N-linked glycosylation of proteins (PMID: 20852264). Rare mutations cause autosomal-recessive disorders of glycosylation, type Iq ((#612379) or Kahrizi syndrome (#612713). |
| CCDC157 | rs11912350 | 0.88 | 85 | 0.05 | Intron SF3A1 | 0.1 | – | Very little is known about the role of the CCDC157 gene, there are no specific publications. Co-localization is observed with multiple other transcripts at this locus. |

PP posterior probability, DHS DNAse I hypersensitivity site, SNP index SNP from the EA-specific meta-analysis
This table includes all genes with high posterior probability (H4 ≥ 0.8) of co-localization of the UACR association signal and gene expression in kidney tissues.
1*: ENCODE kidney, 2* ENCODE epithelial, 3* Roadmap kidney

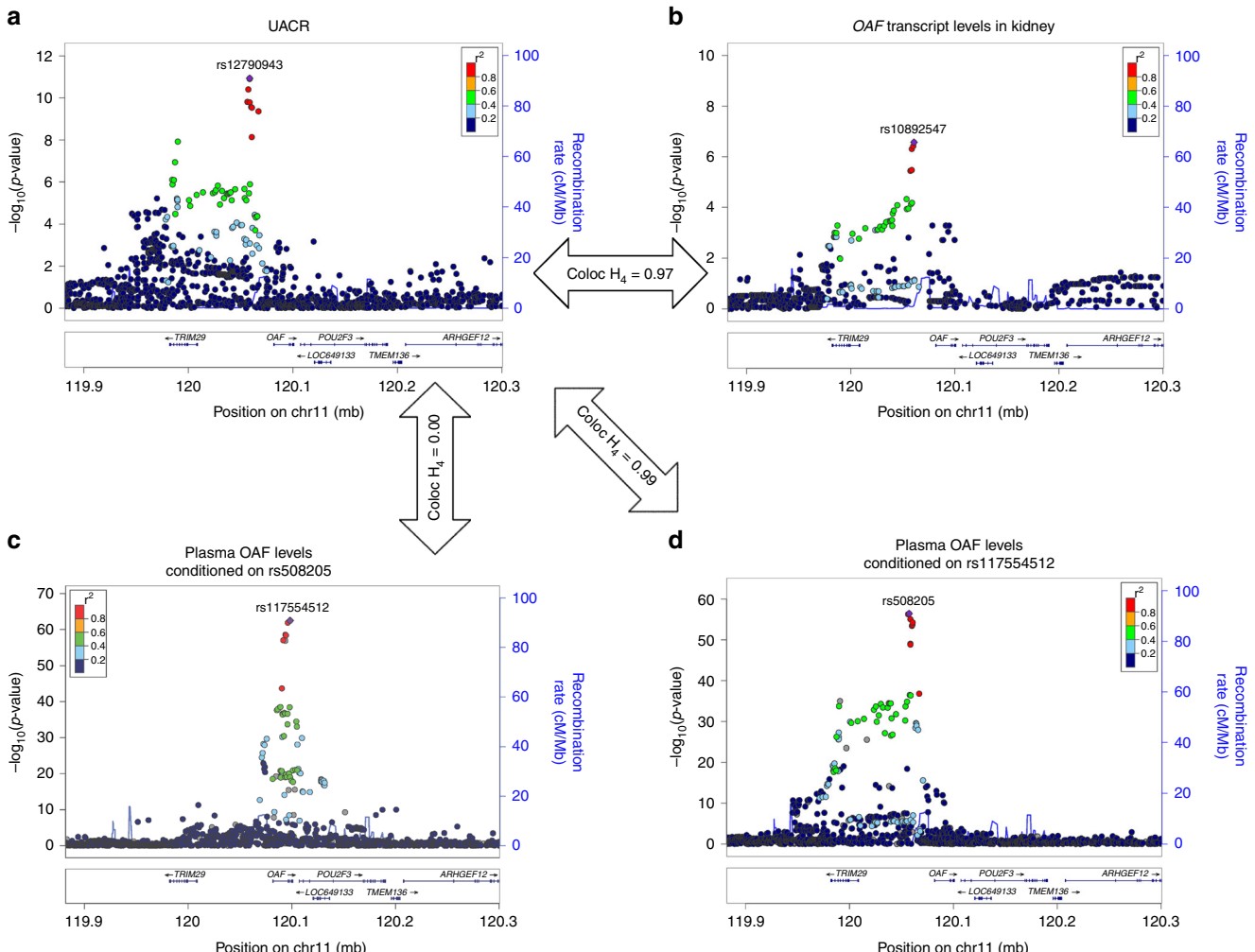

**Fig. 7** Co-localization of association signals of the *OAF* locus. Regional association plots of the *OAF* locus in the European ancestry urinary albumin-to-creatinine ratio (UACR) GWAS (**a**), with *OAF* gene-expression levels in healthy kidney tissue sections (**b**), and with OAF plasma levels (**c**, **d**). The dots are colored according to their correlation $r^2$ with the index SNP estimated based on the 1000 Genomes EUR reference samples (gray for missing data). This locus has two independent pQTLs for OAF levels, where panel **c** shows the association between the index pQTL at the locus (rs117554512) conditioned on its secondary signal (indexed by rs508205), and panel **d** shows the association with a conditionally independent SNP (rs508205, $r^2 < 0.01$ in 1000 Genomes EUR). The secondary signal rs508205 has strong evidence of co-localization with the UACR association signal (posterior probability $H_4 = 0.99$, Methods), while the signal rs117554512 has not (posterior probability $H_4 = 0$). There was strong evidence of co-localization between the UACR association signal and *OAF* expression in kidney tissue (posterior probability $H_4 = 0.97$)

loss of slit diaphragm proteins (Fig. 8g, h; 3D reconstruction Supplementary Fig. 9K). This implies that the polarity factor aPKC is directly involved in slit diaphragm formation, consistent with studies in murine podocytes[53]. Staining patterns were comparable when silencing *oaf* and *aPKC* using second RNAi lines (Supplementary Fig. 9F–I). In summary, the *Drosophila* data support a role of *OAF* in tubular protein endocytosis and *PRKCI* in slit diaphragm formation.

## Discussion

In this GWAS meta-analysis of UACR, we identified 68 loci in total, the majority of which was associated with urinary albumin concentrations and MA. Statistical fine-mapping and co-localization analyses with gene expression across 47 human tissues and with plasma protein levels resolved GWAS loci into novel driver genes and variants. This approach allowed for translating two genes prioritized in our workflow, *OAF* and *PRKCI*, into mechanistic insights in an in vivo experimental model of proteinuria. Genome-wide genetic correlation analyses

and a phenome-wide association study of a genetic risk score for UACR in a large independent population highlighted a common genetic component or co-regulation with traits and diseases with renal, hepatic, or endothelial components. Together, these results represent a comprehensive resource for translational research into albuminuria.

Until recently, GWAS of UACR in mostly population-based studies only identified and replicated two loci: *CUBN*[22,54] and *HBB*[24], detected through an earlier candidate gene study[33]. In addition to these two loci, we also identified the *BCL2L11* locus, reported in an earlier admixture mapping study[25], with the index SNP mapping to the neighboring *ACOXL* gene. Our fine-mapping workflow did not provide strong evidence for either *ACOXL* or *BCL2L11* as the likely causal gene. We did not identify genome-wide significant signals at *RAB38* and *HS6ST1* among persons with diabetes, which we reported in an earlier study at suggestive significance[23]. Potential reasons include differences in quantification and statistical transformation of UACR, different participating studies, and false-positive results in the initial report. Twenty-eight of the 61 loci detected in EA individuals

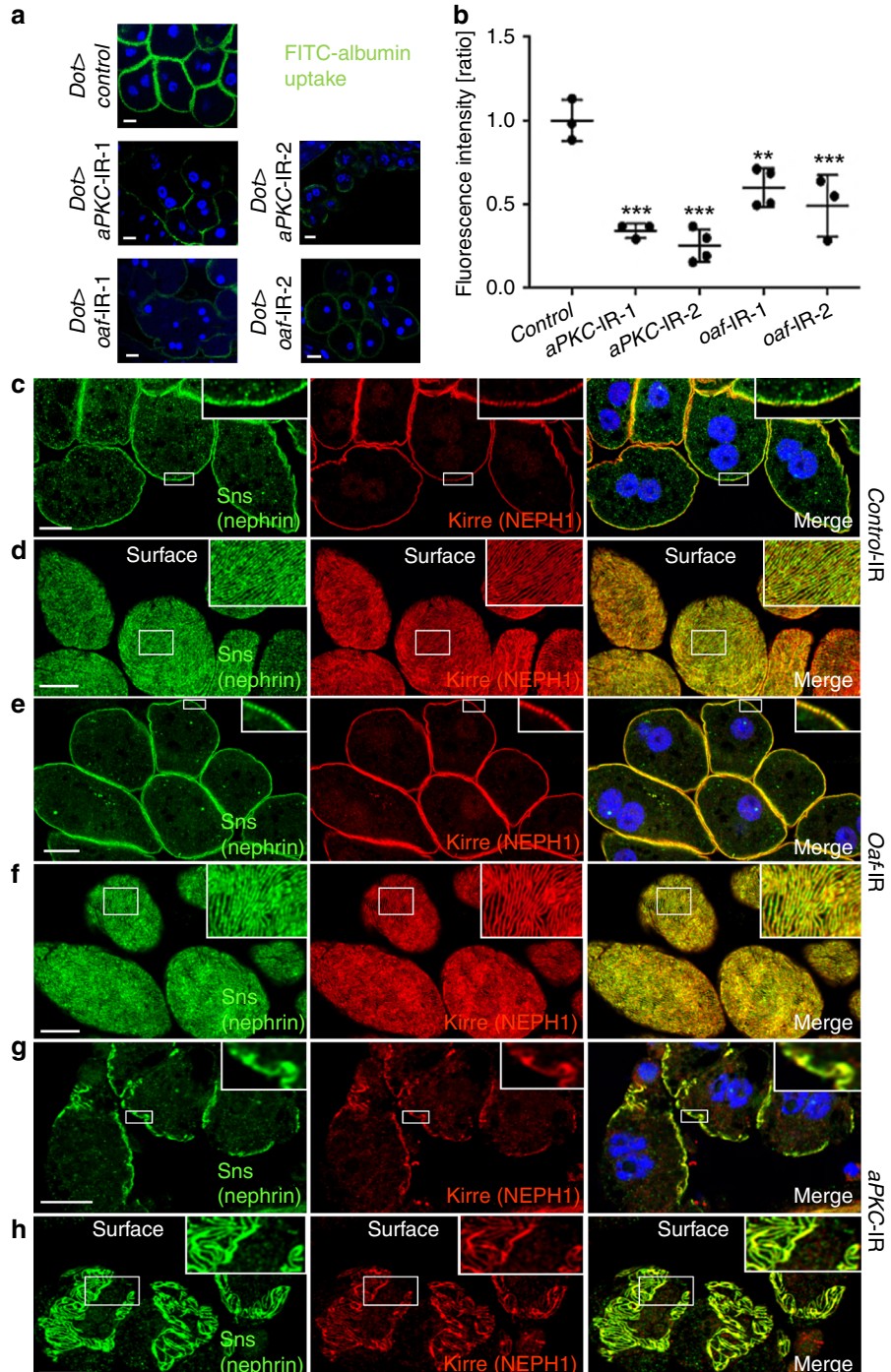

**Fig. 8** In vivo results of *Drosophila* orthologs. The *Drosophila* orthologs of *OAF* and *PRKCI* (*aPKC*) are both required for nephrocyte function and *aPKC*-RNAi affects slit diaphragm formation. **a** Garland cell nephrocytes were exposed to FITC-albumin. Nephrocytes expressing control RNAi exhibit intense endocytosis, while expression of RNAi directed against *oaf* and *aPKC* (ortholog of *PRKCI*) decreases tracer uptake. **b** Quantitation of fluorescence intensity from FITC-albumin uptake is shown for the indicated genotypes. Values are presented as mean ± standard deviation of the ratio to a control experiment. Statistical significance was calculated using ANOVA and Dunnett's post hoc analysis. A statistically significant difference (defined as $p < 0.05$) is observed for *oaf*-RNAi-1 ($N = 4$), *oaf*-RNAi-2 ($N = 3$), *aPKC*-RNAi-1 ($N = 3$), and *aPKC*-RNAi-2 ($N = 4$), where ** indicate $p < 0.01$ and ***$p < 0.001$. **c** Staining the slit diaphragm proteins Sns (ortholog of nephrin) and Kirre (ortholog of NEPH1) in control nephrocytes shows regular formation of slit diaphragms. Airyscan technology partially allows for distinguishing individual slit diaphragms (insets). **d** Tangential sections through the surface of control nephrocytes reveals the regular fingerprint-like pattern of slit diaphragm proteins. **e**, **f** Expression of *oaf*-RNAi-1 does not entail an overt phenotype, suggesting reduced nephrocyte function may be a consequence of impaired protein reabsorption while slit diaphragm formation is not affected. **g**, **h** Expression of *aPKC*-RNAi-1 results in a clustered and irregular pattern of slit diaphragm proteins (insets in **g**) and a complete loss of slit diaphragm protein distinct areas on the cell surface. This suggests the loss of nephrocyte function is a consequence of impaired slit diaphragm formation. All scale bars represent 10 μm

were also reported in the recent Mendelian Randomization study of albuminuria[27], which is not surprising given the inclusion of UKBB data in our meta-analysis. Still, our study identifies 32 additional loci for UACR in the overall sample, as well as four among people with diabetes. Moreover, results allow for prioritization of loci with respect to their association with urinary albumin, whereas previous studies have not evaluated whether UACR-associated loci were driven by associations with urinary albumin, creatinine, or both.

Previous GWAS of albuminuria have not resolved associated loci into underlying genes and variants. Our workflow identified co-localization of UACR-associations with differential gene expression of *PRKCI*, *TGFB1*, *WIPF3*, *PTH1R*, *CYP2S1*, and *MUC1* in glomerular kidney portions and *OAF*, *SRD5A3*, and *CCDC157* in tubulointerstitial tissue. Some of these genes already have established roles in the function of the glomerular filter in diabetic (*TGFB1*)[55,56] and monogenic kidney disease (*MUC1*)[49], while others such as *OAF or WIPF3* represent novel candidates or, as for *PRKCI*, have not yet been implicated in humans[53]. Our combination of human and *Drosophila* studies support a role of *PRKCI* in glomerular filtration function and of *OAF* in tubular protein reabsorption, where reduced endocytosis upon gene silencing reflects the human allele associated with higher UACR and lower *OAF* expression and plasma levels. The lack of a phenotype upon silencing of the *WIPF3* ortholog may reflect the unclear state of orthology, a lack of evolutionary conservation, or potentially an insufficient knockdown.

Several insights from our study are of clinical interest. First, the clinical relevance of genes detected in our screen, *CUBN* and *COL4A4*, is underscored by a respective monogenic disease featuring albuminuria and kidney disease, Imerslund-Grasbeck (MIM 261100) and Alport syndrome (MIM 203780). Second, the identification of *NR3C2*, encoding an essential component of the RAAS, links this pathway to both albuminuria and adverse clinical outcomes. Pharmacological inhibition of the RAAS has been shown to be associated with reduced risk of end-stage kidney disease[12] and cardiovascular events[10,13–15], suggesting that genetic studies of UACR in large human populations may identify pathways amenable to pharmacological intervention that reduce both albuminuria and CVD risk. Third, the genome-wide genetic correlations of UACR and the UACR GRS associations may point toward diseases with a common genetic basis or to co-regulation of disease-relevant cell types. The latter could be reflected in the role of the liver in lipid metabolism and albumin production, the role of the kidney in urate metabolism and albumin excretion, and the role of the endothelium in hypertension and glomerular filtration. A potential role of the endothelium and the vasculature is further corroborated by the significantly enriched pathway "abnormal placental labyrinth vasculature morphology" and many other nominally enriched pathways related to angiogenesis, as well as the identification of the *VEGFA* (Vascular Endothelial Growth Factor A; *LINC01512*) locus, an important growth factor for vascular endothelial cell migration and proliferation. Interestingly, a recent Mendelian Randomization analysis of UACR and blood pressure supported a causal relationship between the two, but reported that SNPs in *CUBN* and *CYP1A1* were only associated with UACR and not blood pressure. We find that the index SNPs in *CUBN* and *CYP1A1* are related to UACR via tubular albumin reabsorption and an association with urinary creatinine but not albumin, respectively. This may indicate that the increased filtration of albumin in the glomerulus, potentially as a result of endothelial damage, and not albuminuria per se may link albuminuria to hypertension and increased CVD risk. Fourth, albuminuria is a hallmark of diabetic kidney disease and associated with unfavorable outcomes. Understanding pathways underlying albuminuria in diabetes may therefore be of particular

relevance, and the four novel diabetes-specific loci identified in our study may represent a first step into this direction. Lastly, translation of GWAS loci into differential plasma protein levels as observed for OAF is of particular interest, as plasma protein levels represent both potential biomarkers and interventional targets.

Strengths of our study include its standardized approach to phenotype definition, its large samples size, internal locus validation, and the study of participants with diabetes. The identification of a previous Amerindian-specific locus[25] in our transethnic analysis underscores the value of studying diverse ancestries, but EA individuals are still strongly overrepresented, which limits the power to detect heterogeneity correlated with ancestry. Limitations that are not specific to our study are related to the accurate quantification of UACR, which is influenced by biologic variation of urinary albumin, by the sensitivity and variation of albumin assays, and by standardization to urinary creatinine to account for urine dilution[23]. We addressed these issues by harmonizing UACR calculation across cohorts, and by separate assessment of associations with urinary albumin and creatinine. Across-cohort variation was overcome to some degree by the use of a central lab in the large UKBB, but may also introduce findings related to UKBB-specific sample handling, storage, measurement, or exposures. The statistical fine-mapping focused on SNPs available in the majority of studies, which might have limited the discovery of novel associations or the fine-mapping of population-specific or low-frequency variants. Such analyses represent avenues for future research. Other fine-mapping methods such as Bayesian approaches that incorporate priors based on variant annotation exist, but ultimately all statistically prioritized variants need to be experimentally validated.

In summary, we identified and characterized 68 loci associated with UACR and highlight potential causal genes, driver variants, target tissues, and pathways. These findings will inform experimental studies and advance the understanding of albuminuria and correlated traits, an essential step for the development of novel therapies to reduce the burden of CKD and potentially CVD.

## Methods
We set up a collaborative meta-analysis based on a distributive data model. An analysis plan was developed and circulated to all participating studies via a Wiki system [https://ckdgen.eurac.edu/mediawiki/index.php/CKDGen_Round_4_EPACTS_analysis_plan]. Phenotypes were generated and quality checks performed within each study in a standardized manner through scripts provided to all study centers. Before conducting the analyses, studies uploaded automatically generated PDF and text files. After approval of the phenotype quality, ancestry-specific GWAS were performed in each study and uploaded centrally. Files were quality controlled using GWAtoolbox[57] and customized scripts, harmonized, and meta-analyzed. Details regarding each step are provided below. Each study was approved by the respective ethics committee, and all participants provided written informed consent. Drosophila research was carried out in compliance with all relevant ethical regulations. Drosophila experiments are exempt from a specific regulatory approval.

**Phenotype definition.** Methods for the measurement of urinary albumin and creatinine in each study are reported in Supplementary Data 1. Urinary albumin values below the detection limit of the used assays were set to the lower limit of detection, and the UACR was assessed in mg/g and calculated as urinary albumin (mg/l)/urinary creatinine (mg/dl) × 100. MA cases were defined as UACR > 30, and controls as UACR < 10 mg/g, no other exclusions were applied. These steps were all included in the distributed phenotyping script. MA GWAS analyses were limited to studies with ≥100 MA cases.

**GWAS in individual studies.** In each study, genotyping was performed using genome-wide arrays followed by application of study-specific quality filters prior to phasing and imputation. Genome-wide data were imputed to the Haplotype Reference Consortium (HRC) version 1.1, 1000 Genomes Project (1000G) phase 3 v5 ALL, or the 1000G phase 1 v3 ALL reference panels using the Sanger [https://imputation.sanger.ac.uk/] and Michigan Imputation Server [https://imputationserver.sph.umich.edu/]. Detailed information on study-specific

genotyping, imputation, and QC is provided in Supplementary Data 2. Unless indicated differently, variants are annotated according to the GRCh37 (hg19) reference build.

The inverse normal transformed age-adjusted and sex-adjusted residuals of log-transformed UACR, as well as urinary albumin and urinary creatinine levels separately for the sensitivity analysis in the UKBB sample, were used as the dependent variable in a linear regression model fitted in each study-specific GWAS. For MA, a logistic regression model adjusted for sex and age was used. The models were adjusted for study-specific covariates, such as recruitment site and genetic principal components where applicable. Family-based studies used mixed-effect models by including the relationship of the individuals as a variance component. Additive genetic models were fitted using the SNP's allele dosage as an independent variable. The analysis programs used for the GWAS are provided in Supplementary Data 2.

**GWAS meta-analysis**. For UACR, studies contributed a total of 54 GWAS summary statistics files. After QC, the total samples size was 564,257 (547,361 individuals of European ancestry [EA], 6324 of East Asian ancestry [EAS], 6795 African Americans [AA], 2335 of South Asian ancestry [SA], and 1442 Hispanics; Supplementary Data 1). For MA, a total of 38 GWAS summary files were contributed, totaling a post-QC samples size of 348,954 (51,861 cases; Supplementary Data 1). Both meta-analyses included individuals with and without diabetes.

Before meta-analysis, study-specific GWAS files were filtered to retain only SNPs with imputation quality (IQ) score > 0.6 and MAC > 10, effective sample size ≥ 100, and a |beta| < 10 to remove implausible outliers. Within study, we estimated the genomic inflation factor $\lambda_{GC}$ and applied GC correction when $\lambda_{GC}$ was >1. Fixed effects inverse-variance weighted meta-analysis of the study-specific GWAS result files was performed using METAL[58], which was adapted to obtain effects and standard errors of higher precision if required (seven decimal places instead of four). After meta-analysis of 37,915,339 SNPs, we retained only variants that were present in ≥50% of the GWAS data files (27 studies) and had a total MAC of ≥400. Across ancestries, this yielded 8,034,757 variants for UACR (8,603,712 in EA with an observed MAF > 0.3%), and 8,326,000 variants for MA.

The inflation of $p$-values attributed to reasons other than polygenicity was assessed by LD score regression.[59] The intercept was estimated as 0.95, and thus ≤1, indicating that any residual inflation was likely due to polygenicity rather than confounding. Therefore, $p$-values were not corrected for a second round of genomic control after the meta-analysis.

The genome-wide significance level was set at $5 \times 10^{-8}$. Between-study heterogeneity was assessed using the $I^2$ statistic.[60] Variants were assigned to loci by selecting the SNP with the lowest $p$-value genome-wide as the index SNP, defining the corresponding locus as the ±500 kb region around it, and repeating the procedure until no further genome-wide significant SNP remained. A locus was considered novel if it did not contain any variant identified by previous GWAS of UACR. The loci were named according to the nearest gene of the index SNP, the SNP with the lowest $p$-value within a locus.

For UACR, we evaluated heterogeneity correlated with ancestry using study-specific GWAS files filtered for polymorphic SNPs with an IQ score > 0.3, an effective sample size ≥ 100, and a |beta| < 10. Analysis was performed using the software Meta-Regression of Multi-Ethnic Genetic Association (MR-MEGA v0.1.2.25)[28], where the meta-regression model included the three axes explaining the largest genetic variation estimated from allele frequencies provided in the study-specific GWAS files.

The narrow-sense heritability of the trait based on all SNPs with a MAF > 1% was estimated using the genome-wide summary statistics for UACR with the MHC region removed as input for the LD score regression software[59], using the 1000 Genomes phase 3 EUR reference panel for estimating LD. The proportion of phenotypic variance explained by the index SNPs was estimated as $\beta^2 * 2 * MAF * (1-MAF)$, with $\beta$ representing the SNP effect and accounting for a trait variance of 1 due to the inverse normal transformation of the analyzed trait. Thus, the estimates provide the proportion of the variance of sex- and age-adjusted log-transformed UACR that is explained by the respective SNPs. The expected number of discoveries in future, larger studies and the corresponding percentage of GWAS heritability explained with increases in sample size was estimated using a recently published method[31]. The summary statistics of the UACR trans-ethnic meta-analysis were used as input.

**Functional enrichment**. We used DEPICT[36] version 1 release 194 to identify gene sets and tissue/cell types enriched in UACR-associated loci. DEPICT performs gene set and tissue-/cell-type enrichment analysis by testing whether genes in GWAS-associated loci are enriched in 14,461 reconstituted gene sets. These reconstituted gene sets were generated based on a large number of predefined gene sets from diverse molecular pathway databases including protein–protein interactions, and gene sets from mouse gene knockout studies. The function of each gene in 14,461 reconstituted gene sets was predicted from co-regulation analyses of 77,840 expression microarray samples. Tissues and cell-type enrichment was conducted in DEPICT by testing whether the genes in associated regions were highly expressed in any of 209 MeSH annotations for 37,427 microarrays. We included all variants that reached a genome-wide significant $p$-value of association with UACR ($p < 5 \times 10^{-8}$) from the trans-ethnic meta-analysis. DEPICT analysis was conducted with

500 repetitions to compute FDR and 5000 permutations to compute enrichment test $p$-values adjusted for gene length by using 500 null GWAS.

**Phenome-wide association study**. All analyses were conducted using standard PheWAS coding methodologies[37] using the R-package "PheWAS". Models were adjusted for ten genetic principal components and sex, when appropriate. All analyses were conducted among 192,868 participants of European ancestry in the Million Veteran Program sample. A weighted genetic risk score was first built using the 59 UACR-associated SNPs (Supplementary Data 3) where the UACR-increasing allele was coded as the effect allele. Based on the number of covariates included in the model, only traits with ≥100 cases were included in the analysis resulting in evaluation of 1422 traits. A Bonferroni threshold of $3.5 \times 10^{-5}$ (0.05/1422) was applied for assessing significance of the association test.

The genetic UACR risk score was also tested for association with additional outcomes using GWAS summary statistics with association testing implemented in the function $grs.summary()$ of the R-package "gtx". The summary statistics for hypertension and heart failure were calculated in the UKBB prior to the risk score association analysis. Hypertension cases were defined based on ICD-10 codes (I10, I11, I11.0, I11.9, I12, I12.0, I12.9, I13, I13.0, I13.1, I13.2, I15.0, I15.1, I15.2, I15.8, and I15.9), as self-reported hypertension or essential hypertension, by measured systolic blood pressure ≥ 140 mmHg, diastolic blood pressure ≥ 90 mmHg, or by taking blood pressure medication. Hear failure cases were defined based on ICD-10 codes (I11.0, I13.0, I13.2, I25.5, I42.0, I42.5, I42.8, I42.9, I50, I50.0, I50.1, and I50.9), or by self-reported cardiomyopathy, excluding hypertrophic cardiomyopathy. The summary statistics for other outcomes were based on results from published GWAS meta-analyses with references provided in Supplementary Table 2. Statistical significance was defined as $p < 0.007$ of the association test after correction for the number of evaluated associations (0.05/7).

**Genetic correlation with other traits**. Genome-wide genetic correlations between UACR and UK Biobank traits and diseases were evaluated to investigate whether there was evidence of co-regulation or a shared genetic basis, both known and novel. Using LD score regression that can account for overlapping samples[61] and the EA association summary statistics as input, we evaluated pair-wise genetic correlations between UACR and each of 517 pre-computed GWAS summary statistics of UKBB traits and diseases available through the web-platform LDHub. An overview of the sources of these summary statistics and their corresponding sample sizes is available at [http://ldsc.broadinstitute.org]. Statistical significance was assessed at the Bonferroni corrected level of $9.7 \times 10^{-5}$ (0.05/517).

**Second signals within identified loci**. To identify additional, independent UACR-associated variants within the identified loci, approximate conditional analyses were carried out that incorporated LD information from an ancestry-matched reference population. We used the genome-wide UACR summary statistics from the EA meta-analysis as input, because an LD reference sample scaled to the size of our meta-analysis was only available for EA individuals[43,44]. We randomly selected 15,000 participants from the UK Biobank data set (UKBB; application ID 2027, data set ID 8974). Individuals who withdrew consent and those not meeting data cleaning requirements were excluded, keeping only those who passed sex check, had a genotyping call rate of ≥95%, and did not represent outliers with respect to SNP heterozygosity. For each pair of individuals, the proportion of variants shared identical-by-descent (IBD) was computed using PLINK [https://www.cog-genomics.org/plink/]. We retained only one member of each pair with an IBD coefficient of ≥0.1875. Individuals were restricted to those of EA by excluding outliers along the first two PCs from a principal component analysis using the HapMap phase 3 release 2 populations as reference. The final data set to estimate LD included 13,558 EA individuals and 16,969,363 SNPs.

Basis for statistical fine-mapping were the 61 1-Mb genome-wide significant loci identified in the EA meta-analysis, clipping at chromosome borders. Overlapping loci as well as pairs of loci whose respective index SNPs were correlated ($r^2 > 0.1$ in the UKBB data set described above) were merged, resulting in a final list of 57 regions prior to fine-mapping. Within each region, the GCTA stepwise model selection procedure (cojo-slct algorithm) was used to identify independent variants employing a stepwise forward selection approach[44]. We used the default collinearity cutoff of 0.9 and set the significance threshold to identify independent SNPs to $5 \times 10^{-8}$.

**Estimation of credible sets**. Statistical fine-mapping was carried out for each of the 57 merged regions used as input for GCTA cojo-slct. For each region that contained multiple independent SNPs identified by the GCTA stepwise forward selection approach, approximate conditional analyses conditioned on all remaining independent SNP of this region were carried out using the GCTA cojo-cond algorithm to estimate conditional effect sizes. The derived effect estimates were used in the Wakefield's formula as implemented in the R-package 'gtx' version 2.0.1 [https://github.com/tobyjohnson/gtx] to derive approximate Bayes factors (ABF) from conditional estimates in regions with multiple independent SNPs, and from the original estimates for regions with a single independent SNP. Given that 95% of the SNP effects from the UACR GWAS were within ±0.03, the standard deviation prior was chosen as 0.0153 based on formula (8) in the original publication[45]. For

each variant within an evaluated region, the Approximate Bayes Factor obtained from the effect and its standard error of the marginal (single signal region) or conditional estimates (multi-signal regions) was used to calculate the PP for the variant driving the association signal (causal variant). For each region, 99% credible sets, representing the set of SNPs that contain with a 99% PP the variant causing the association, were calculated by summing up the PP-ranked variants until the cumulative PP was >99%.

**Functional annotation of identified variants.** Functional annotations of index variants of associated loci and credible set variants were performed by querying the SNiPA database v3.2 (March 2017) [https://snipa.helmholtz-muenchen.de/snipa/]. SNiPA includes extensive annotations ranging from regulatory elements, over gene annotations to variant annotations and published GWAS associations. SNiPA release v3.2 is based on 1000 Genomes phase 3 version 5 and Ensembl version 87 data sets. The Ensembl VEP tool [https://www.ensembl.org/info/docs/tools/vep/] was used for primary effect prediction of SNPs. The CADD score[62] provided by SNiPA is based on CADD release v1.3 and presented as PHRED-like transformation of the C score.

**Co-localization of UACR and cis-eQTL associations.** Co-localization analysis was based on the genetic associations with UACR in the EA sample (because the great majority of gene expression data sets was generated from EA). Gene expression was quantified from microdissected human glomerular and tubulointerstitial kidney portions from 187 individuals participating in the NEPTUNE study[48], as well as from the 44 tissues included in the GTEx Project version 6p release [https://gtexportal.org/]. The eQTL and GWAS effect alleles were harmonized. For each locus, we identified tissue–gene pairs with reported eQTL data within ±100 kb of each GWAS index variant. The region for each co-localization test was defined as the eQTL cis window defined in the underlying GTEx and NephQTL studies. We used the default parameters and prior definitions set in the "coloc.fast" function from the R-package "gtx" version 2.0.1 [https://github.com/tobyjohnson/gtx], which is an adapted implementation of Giambartolomei's co-localization method[63]. The same package was also used to estimate the direction of effect as the ratio of the average PP (that was obtained from credible set estimation) weighted GWAS effects over the PP weighted eQTL effects.

An additional co-localization analysis was performed using a complementary gene-expression data set derived from healthy human kidney tissue. The corresponding eQTL data set was generated by correlating genotype with RNA-seq-based gene expression levels from 96 human kidney samples[47]. Co-localization analysis of GWAS signals and eQTL signals was performed using Coloc[63], using the same distance criteria to identify shared eQTL and GWAS regions as above, including variants within the cis-window (±1 Mb from TSS) of each identified candidate gene, and the parameters $p1 = 1 \times 10^{-4}$, $p2 = 1 \times 10^{-4}$, and $p12 = 1 \times 10^{-5}$.

For all co-localization analyses, a PP ≥ 0.8 of the H4 test (one common causal variant underlying UACR and eQTL association signal) was applied to select a significant result.

**Trans-eQTL analysis.** We performed trans-eQTL annotation through LD mapping based on the 1000 Genomes phase 3 version 5 European reference panel with a $r^2$ cutoff of >0.8. We limited annotation to index SNPs with a fine-mapping PP ≥1% in at least one fine-mapped-region. Due to expected small effect sizes, only available genome-wide trans-eQTL studies of either peripheral blood mononuclear cells or whole blood with a sample size of ≥1000 individuals were considered, resulting in five non-overlapping studies[64–68]. For the study by Kirsten et al.[68], we had access to an update with larger sample size combining two nonoverlapping studies (LIFE-Heart and LIFE-Adult) resulting in a total sample size of 6645. To improve stringency of results, we focused the analysis on inter-chromosomal trans-eQTLs with association test p-values of $p < 5 \times 10^{-8}$ reported by ≥2 studies (Supplementary Table 4).

**pQTL lookup and co-localization.** The pQTL data were generated using an aptamer-based multiplex protein assay (SOMAscan) to quantify 3622 proteins from stored EDTA plasma of 3301 healthy participants of the INTERVAL study, which were genotyped on the Affymetrix Axiom UK Biobank genotyping array and imputed to a combined 1000 Genomes Phase 3-UK10K reference panel[50]. For this lookup, all pQTLs with $p < 1 \times 10^{-4}$ were selected.

Co-localization analysis for pQTL data was performed using the same analysis approach as described for eQTL co-localization. For associations with plasma protein concentrations, pQTL results of 1927 genetic associations with 1478 proteins obtained by the Somalogic proteomics platform GWAS[50] were included. In a first instance, pQTLs within a ± 500 kb region of each UACR-associated SNP (Supplementary Data 5) were identified. In case a pQTL region contained multiple independent index SNPs, additional pQTLs were calculated conditioning on the respective index SNP. Next, the conditional and unconditional pQTLs ($n = 38$) were included in the co-localization analysis using the coloc.abf() function with default priors of the R-package "coloc" implementing the co-localization method of Giambartolomei[63].

The intra-assay coefficient of variation for the OAF protein, for which evidence for co-localization of the UACR association and OAF plasma levels was identified, was 5.7% and 16.9% in the two batches of SOMAscan measurements[50].

**Drosophila experiments.** Transgenic RNAi studies were performed using the UAS/GAL4 system, flies were raised on standard agar cornmeal molasses. RNAi crosses were grown at 30 °C. The RNAi stocks were obtained from the Bloomington Drosophila Stock Center at Indiana University (oaf-RNAi-1 BDSC #40926, aPKC-RNAi-1 BDSC # 35001, aPKC-RNAi-2 BDSC #34332) or the Vienna Drosophila Resource Center respectively (oaf-RNAi-2 VDRC #38257, Vrp1-RNAi-1 VDRC #102253, Vrp1-RNAi-2 VDRC #23888). Control RNAi was directed against EGFP (BSDC# 41553). Dorothy-GAL4 (BDSC #6903) was used to drive expression in nephrocytes.

To perform the FITC-albumin endocytosis assay, garland cell nephrocytes were dissected from wandering third instar larvae in PBS and incubated with 0.2 mg/ml FITC-albumin (Sigma) for 30 s. Cells were rinsed briefly with ice-cold PBS four times and fixed immediately for 5 min in 8% paraformaldehyde in presence of Hoechst 33342 (1:1000). Cells were mounted in Roti-Mount FluorCare (Carl Roth GmbH) and imaged using a Zeiss LSM 880 confocal microscope. Quantification of fluorescent tracer uptake was performed with ImageJ software. Average fluorescence of the three brightest cells was measured and intensity of the background subtracted. The results are expressed as a ratio to a control experiment with EGFP-RNAi that was performed in parallel.

For immunohistochemistry, garland cell nephrocytes were dissected from wandering third instar larvae, fixed for 20 min in PBS containing 4% paraformaldehyde, and stained according to the standard procedure. The following primary antibodies were used: rabbit anti-sns (1:500, gift from S. Abmayr), guinea pig anti-Kirre (1:200, gift from S. Abmayr), and rabbit anti anti-PKCζ (C20) (1:200, sc-216-G, Santa Cruz Biotechnology) that was previously shown to detect Drosophila aPKC[69]. For imaging, a Zeiss LSM 880 confocal microscope was used. Image processing was done by ImageJ and Gimp software. Three-dimensional reconstruction of confocal images was done using Imaris software.

**Reporting summary.** Further information on research design is available in the Nature Research Reporting Summary linked to this article.

## Data availability

Summary genetic association results are freely available on the CKDGen Consortium website [https://ckdgen.imbi.uni-freiburg.de/]. The source data underlying Figs. 1, 2, 5–8 and Supplementary Figs. 8 and 9 are provided as a Source Data file. The source data underlying Figs. 3, 4, and Supplementary Fig. 7 are provided in Supplementary Data 9, 10, and 7, respectively, and the data underlying the Supplementary Figs. 2–6 are based on the respective downloadable summary genetic association results.

## Code availability

The script for generating the phenotypes used in the GWAS is available via GitHub [https://github.com/genepi-freiburg/ckdgen-pheno].

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

## Acknowledgements

We thank Daniele Di Domizio (Eurac Research) and Jochen Knaus (University of Freiburg) for IT assistance, Toby Johnson (GSK) for sharing his code and helpful discussions related to the fine-mapping and co-localization analyses, and Roland Nitschke, Life Imaging Centre, University of Freiburg, for help with confocal microscopy. We thank Lina L. Kampf for technical assistance with the *Drosophila* experiments and Jan Halbritter for work related to the recruitment of the Sorbs cohort. This research has been conducted using the UK Biobank Resource under application numbers 20272, 10019, and 31852. Study-specific acknowledgements, consortium members, and funding sources are listed in the Supplementary Note 1.

## Author contributions

Manuscript writing group: Alexander Teumer, Yong Li, Sahar Ghasemi, Bram P. Prins, Matthias Wuttke, Tobias Hermle, Nisha Bansal, Harold Snieder, Adam S. Butterworth, Adriana M. Hung, Cristian Pattaro, Anna Köttgen. Design of this study: Alexander Teumer, Matthias Wuttke, Tobias Hermle, Karsten B. Sieber, Adrienne Tin, Mathias Gorski, Christian Fuchsberger, Carsten A. Böger, Andrew P. Morris, Markus Scholz, Adam S. Butterworth, Cristian Pattaro, Anna Köttgen. Bioinformatics: Shreeram Akilesh, Daniela Baptista, Carsten A. Böger, Robert J. Carroll, Audrey Y. Chu, Massimiliano Cocca, Tanguy Corre, Frauke Degenhardt, Jasmin Divers, Georg Ehret, Andre Franke, Sahar Ghasemi, Ayush Giri, Mathias Gorski, Franziska Grundner-Culemann, Pavel Hamet, Iris M. Heid, Anselm Hoppmann, Katrin Horn, Johanna Jakobsdottir, Navya Shilpa Josyula, Chiea-Chuen Khor, Holger Kirsten, Anna Köttgen, Carl D. Langefeld, Man Li, Yong Li, Jianjun Liu, Leo-Pekka Lyytikäinen, Jonathan Marten, Dennis O. Mook-Kanamori, Peter J. van der Most, Raymond Noordam, Teresa Nutile, Sarah A. Pendergrass, Anna I. Podgornaia, Chengxiang Qiu, Markus Scholz, Sanaz Sedaghat, Christian M. Shaffer, Karsten B. Sieber, Albert V. Smith, Silke Szymczak, Alexander Teumer, Hauke Thomsen, Johanne Tremblay, Chaolong Wang, Matthias Wuttke, Yizhe Xu, Zhi Yu. Functional analysis of candidate genes in *Drosophila*: Tobias Hermle, Mengmeng Chen, Lea Gerstner. Genotyping: Daniela Baptista, Ralph Burkhardt, Carsten A. Böger, Ching-Yu Cheng, Georg Ehret, Mary F. Feitosa, Andre Franke, Christian Fuchsberger, Ron T. Gansevoort, Pavel Hamet, Pim van der Harst, Erik Ingelsson, Chiea-Chuen Khor, Wolfgang Koenig, Peter Kovacs, Florian Kronenberg, Mika Kähönen, Antje Körner, Leslie A. Lange, Terho Lehtimäki, Leo-Pekka Lyytikäinen, Thomas Meitinger, Dennis O. Mook-Kanamori, Andrew P. Morris, Josyf C. Mychaleckyj, Martina Müller-Nurasyid, Nicholette D. Palmer, Dermot F. Reilly, Fernando Rivadeneira, Jerome I. Rotter, Kent D. Taylor, Alexander Teumer, Hauke Thomsen, Johanne Tremblay, André G. Uitterlinden, Uwe Völker, Melanie Waldenberger, Chaolong Wang, Lihua Wang, James G. Wilson, Johan Ärnlöv. Interpretation of Results: Adam S. Butterworth, Carsten A. Böger, Ching-Yu Cheng, Katalin Dittrich, Jasmin Divers, Karlhans Endlich, Mary F. Feitosa, Janine F. Felix, Barry I. Freedman, Sahar Ghasemi, Ayush Giri, Mathias Gorski, Pavel Hamet, Pim van der Harst, Iris M. Heid, Kevin Ho, Katrin Horn, Shih-Jen Hwang, Bettina Jung, Holger Kirsten, Wolfgang Koenig, Anna Köttgen, Carl D. Langefeld, Man Li, Yong Li, Jonathan Marten, Kozeta Miliku, Andrew P. Morris, Nicholette D. Palmer, Cristian Pattaro, Sarah A. Pendergrass, Bram P. Prins, Dermot F. Reilly, Myriam Rheinberger, Markus Scholz, Sanaz Sedaghat, Karsten B. Sieber, Bamidele O. Tayo, Alexander Teumer, Hauke Thomsen, Adrienne Tin, Johanne Tremblay, André G. Uitterlinden, Niek Verweij, Suzanne Vogelezang, Matthias Wuttke, Yizhe Xu, Masayuki Yasuda. Management of an individual contributing study: Shreeram Akilesh, Stephan J.L. Bakker, Murielle Bochud, Eric Boerwinkle, Martin H. de Borst, Hermann Brenner, Adam S. Butterworth, Carsten A. Böger, Robert J. Carroll, Ching-Yu Cheng, Josef Coresh, John Danesh, Olivier Devuyst, Katalin Dittrich, Kai-Uwe Eckardt, Georg Ehret, Janine F. Felix, Oscar H. Franco, Barry I. Freedman, Ron T. Gansevoort, Vilmantas Giedraitis, Alessandro De Grandi, Vilmundur Gudnason, Tamara B. Harris, Pim van der Harst, Andrew A. Hicks, Kevin Ho, Adriana M. Hung, M. Arfan Ikram, Erik Ingelsson, Vincent W.V. Jaddoe, Bettina Jung, Chiea-Chuen Khor, Wieland Kiess, Wolfgang Koenig, Holly Kramer, Florian Kronenberg, Bernhard K. Krämer, Mika Kähönen, Antje Körner, Anna Köttgen, Terho Lehtimäki, Yong Li, Wolfgang Lieb, Su-Chi Lim, Markus Loeffler, Deborah Mascalzoni, Barbara McMullen, Andrew P. Morris, Renée de Mutsert, Jeffrey O'Connell, Afshin Parsa, Cristian Pattaro, Sarah A. Pendergrass, Annette Peters, Belen Ponte, Peter P. Pramstaller, Bruce M. Psaty, Ton J. Rabelink, Dermot F. Reilly, Rainer Rettig, Myriam Rheinberger, Heiko Runz, Charumathi Sabanayagam, Kai-Uwe Saum, Markus Scholz, Ben Schöttker, Harold Snieder, Kari Stefansson, Konstantin Strauch, Michael Stumvoll, Gardar Sveinbjornsson, E-Shyong Tai, Bamidele O. Tayo, Yih-Chung Tham, Joachim Thiery, Adrienne Tin, Johanne Tremblay, Anke Tönjes, Aiko P.J. de Vries, Uwe Völker, James G. Wilson, Otis D. Wilson, Charlene Wong, Tien-Yin Wong, Matthias Wuttke, Qiong Yang, Masayuki Yasuda. Subject Recruitment: Shreeram Akilesh, Hermann Brenner, Carsten A. Böger, Miao-Ling Chee, Katalin Dittrich, Valencia Hui Xian Foo, Barry I. Freedman, Ron T. Gansevoort, Vilmundur Gudnason, Vincent W.V. Jaddoe, Bettina Jung, Florian Kronenberg, Mika Kähönen, Anna Köttgen, Jeannette Jen-Mai Lee, Terho Lehtimäki, Wolfgang Lieb, Lars Lind, Christa Meisinger, Renée de Mutsert, Kjell Nikus, Isleifur Olafsson, Cristian Pattaro, Sarah A. Pendergrass, Belen Ponte, Tanja Poulain, Ton J. Rabelink, Rainer Rettig, Myriam Rheinberger, Nicholas Y. Q. Tan, Andrej Teren, Yih-Chung Tham, Johanne Tremblay, Anke Tönjes, Suzanne Vogelezang, Aiko P.J. de Vries, James G. Wilson, Johan Ärnlöv. Statistical Methods and Analysis: Mary L. Biggs, Carsten A. Böger, Robert J. Carroll, Jin-Fang Chai, Miao-Li Chee, Audrey Y. Chu, Massimiliano Cocca, James P. Cook, Tanguy Corre, Jasmin Divers, Todd L. Edwards, Mary F. Feitosa, Janine F. Felix, Barry I. Freedman, Sandra Freitag-Wolf, Christian Fuchsberger, Sahar Ghasemi, Ayush Giri, Mathias Gorski, Daniel F. Gudbjartsson, Martin Gögele, Pavel Hamet, Pim van der Harst, Iris M. Heid, Anselm Hoppmann, Katrin Horn, Shih-Jen Hwang, Johanna Jakobsdottir, Navya Shilpa Josyula, Bettina Jung, Chiea-Chuen Khor, Holger Kirsten, Holly Kramer, Anna Köttgen, Leslie A. Lange, Carl D. Langefeld, Man Li, Yong Li, Jianjun Liu, Leo-Pekka Lyytikäinen, Anubha Mahajan, Joseph C. Maranville, Jonathan Marten, Kozeta Miliku, Andrew P. Morris, Peter J. van der Most, Matthias Nauck, Boting Ning, Damia Noce, Raymond Noordam, Teresa Nutile, Cristian Pattaro, Sarah A. Pendergrass, Bram P. Prins, Laura M. Raffield, Myriam Rheinberger, Kenneth M. Rice, Fernando Rivadeneira, Kathleen A. Ryan, Markus Scholz, Sanaz Sedaghat, Yuan Shi, Karsten B. Sieber, Albert V. Smith, Benjamin B. Sun, Katalin Susztak, Gardar Sveinbjornsson, Silke Szymczak, Bamidele O. Tayo, Alexander Teumer, Chris H.L. Thio, Hauke Thomsen, Johanne Tremblay, Niek Verweij, Suzanne Vogelezang, Chaolong Wang, Lihua Wang, Matthias Wuttke, Yizhe Xu, Qiong Yang. Critical review of manuscript: all authors.

## Additional information

**Competing interests:** Karsten B. Sieber is full-time employee of GlaxoSmithKline. Gardar Sveinbjornsson, Daniel F. Gudbjartsson, Hilma Holm, Unnur Thorsteinsdottir and Kari Stefansson are full-time employees of deCODE genetics, Amgen Inc. John Danesh is member of the Novartis Cardiovascular and Metabolic Advisory Board, received grant support from Novartis. Oscar H. Franco works in ErasmusAGE, a center for aging research across the life course funded by Nestlé Nutrition (Nestec Ltd.), Metagenics Inc., and AXA. Wolfgang Koenig received modest consultation fees for advisory board meetings from Amgen, DalCor, Kowa, Novartis, Pfizer and Sanofi, and modest personal fees for lectures from Amgen, AstraZeneca, Novartis, Pfizer and Sanofi. Anna I. Podgornaia and Dermot F. Reilly are employees of Merck Sharp Dohme Corp., Whitehouse Station, NJ, USA. Kevin Ho disclosed a research and financial relationship with Sanofi-Genzyme. Bruce M. Psaty serves on the DSMB of a clinical trial funded by the manufacturer (Zoll LifeCor) and on the Steering Committee of the Yale Open Data Access Project funded by Johnson & Johnson. Markus Scholz: Consultancy of and grant support from Merck Serono not related to this project. Adam S. Butterworth received grants from MSD, Pfizer, Novartis, Biogen and Bioverativ and personal fees from Novartis. Anna Köttgen received grant support from Gruenenthal not related to this project. The other authors declare no competing interests.

Alexander Teumer[1,2,160], Yong Li[3,160], Sahar Ghasemi[1,2,160], Bram P. Prins[4,160], Matthias Wuttke[3,5,160], Tobias Hermle[5,160], Ayush Giri[6,7], Karsten B. Sieber[8], Chengxiang Qiu[9], Holger Kirsten[10,11], Adrienne Tin[12,13], Audrey Y. Chu[14], Nisha Bansal[15,16], Mary F. Feitosa[17], Lihua Wang[17], Jin-Fang Chai[18], Massimiliano Cocca[19], Christian Fuchsberger[20], Mathias Gorski[21,22], Anselm Hoppmann[3], Katrin Horn[10,11], Man Li[23], Jonathan Marten[24], Damia Noce[20], Teresa Nutile[25], Sanaz Sedaghat[26], Gardar Sveinbjornsson[27], Bamidele O. Tayo[28], Peter J. van der Most[29], Yizhe Xu[23], Zhi Yu[12,30], Lea Gerstner[5], Johan Ärnlöv[31,32], Stephan J.L. Bakker[33], Daniela Baptista[34], Mary L. Biggs[35,36], Eric Boerwinkle[37], Hermann Brenner[38,39], Ralph Burkhardt[11,40,41], Robert J. Carroll[42], Miao-Li Chee[43], Miao-Ling Chee[43], Mengmeng Chen[5], Ching-Yu Cheng[43,44,45], James P. Cook[46], Josef Coresh[12], Tanguy Corre[47,48,49], John Danesh[50], Martin H. de Borst[33], Alessandro De Grandi[20], Renée de Mutsert[51], Aiko P.J. de Vries[52], Frauke Degenhardt[53], Katalin Dittrich[54,55], Jasmin Divers[56], Kai-Uwe Eckardt[57,58], Georg Ehret[34], Karlhans Endlich[2,59], Janine F. Felix[26,60,61], Oscar H. Franco[26,62], Andre Franke[53], Barry I. Freedman[63], Sandra Freitag-Wolf[64], Ron T. Gansevoort[33], Vilmantas Giedraitis[65], Martin Gögele[20], Franziska Grundner-Culemann[3], Daniel F. Gudbjartsson[27], Vilmundur Gudnason[66,67], Pavel Hamet[68,69], Tamara B. Harris[70], Andrew A. Hicks[20], Hilma Holm[27], Valencia Hui Xian Foo[43], Shih-Jen Hwang[71,72], M. Arfan Ikram[26], Erik Ingelsson[73,74,75,76], Vincent W.V. Jaddoe[26,60,61], Johanna Jakobsdottir[77,78], Navya Shilpa Josyula[79], Bettina Jung[21], Mika Kähönen[80,81], Chiea-Chuen Khor[43,82], Wieland Kiess[11,54,55], Wolfgang Koenig[83,84,85], Antje Körner[11,54,55], Peter Kovacs[86], Holly Kramer[28,87], Bernhard K. Krämer[88], Florian Kronenberg[89], Leslie A. Lange[90], Carl D. Langefeld[56], Jeannette Jen-Mai Lee[18], Terho Lehtimäki[91,92], Wolfgang Lieb[93], Su-Chi Lim[18,94], Lars Lind[95], Cecilia M. Lindgren[96,97], Jianjun Liu[82,98], Markus Loeffler[10,11], Leo-Pekka Lyytikäinen[91,92], Anubha Mahajan[99,100], Joseph C. Maranville[101,158], Deborah Mascalzoni[20], Barbara McMullen[102], Christa Meisinger[103,104], Thomas Meitinger[84,105,106], Kozeta Miliku[26,60,61], Dennis O. Mook-Kanamori[51,107], Martina Müller-Nurasyid[108,109,110], Josyf C. Mychaleckyj[111], Matthias Nauck[2,112], Kjell Nikus[113,114], Boting Ning[115], Raymond Noordam[116], Jeffrey O' Connell[117], Isleifur Olafsson[118], Nicholette D. Palmer[119], Annette Peters[84,120,121], Anna I. Podgornaia[14], Belen Ponte[122], Tanja Poulain[11], Peter P. Pramstaller[20], Ton J. Rabelink[52,123], Laura M. Raffield[124], Dermot F. Reilly[14], Rainer Rettig[125], Myriam Rheinberger[21], Kenneth M. Rice[36], Fernando Rivadeneira[26,126], Heiko Runz[101,159], Kathleen A. Ryan[127], Charumathi Sabanayagam[43,44], Kai-Uwe Saum[38], Ben Schöttker[38,39], Christian M. Shaffer[42], Yuan Shi[43,44], Albert V. Smith[67], Konstantin Strauch[108,109], Michael Stumvoll[128], Benjamin B. Sun[4], Silke Szymczak[64], E-Shyong Tai[18,98,129], Nicholas Y.Q. Tan[43], Kent D. Taylor[130], Andrej Teren[11,131], Yih-Chung Tham[43], Joachim Thiery[11,40], Chris H.L. Thio[29], Hauke Thomsen[132], Unnur Thorsteinsdottir[27], Anke Tönjes[128], Johanne Tremblay[68,133], André G. Uitterlinden[126], Pim van der Harst[134,135,136], Niek Verweij[134], Suzanne Vogelezang[26,60,61], Uwe Völker[2,137], Melanie Waldenberger[84,120,138], Chaolong Wang[82,139], Otis D. Wilson[140], Charlene Wong[45], Tien-Yin Wong[43,44,45], Qiong Yang[115], Masayuki Yasuda[43,141], Shreeram Akilesh[16,142], Murielle Bochud[47], Carsten A. Böger[21,143], Olivier Devuyst[144], Todd L. Edwards[145,146], Kevin Ho[147,148], Andrew P. Morris[46,99], Afshin Parsa[149,150], Sarah A. Pendergrass[151], Bruce M. Psaty[152,153], Jerome I. Rotter[130,154,155], Kari Stefansson[27], James G. Wilson[156], Katalin Susztak[9], Harold Snieder[29],

Iris M. Heid[22], Markus Scholz [10,11], Adam S. Butterworth [4,157,161], Adriana M. Hung [140,146,161], Cristian Pattaro [20,161] & Anna Köttgen [3,12,161]

[1]Institute for Community Medicine, University Medicine Greifswald, Greifswald, Germany. [2]DZHK (German Center for Cardiovascular Research), Partner Site Greifswald, Greifswald, Germany. [3]Institute of Genetic Epidemiology, Department of Biometry, Epidemiology and Medical Bioinformatics, Faculty of Medicine and Medical Center - University of Freiburg, Freiburg, Germany. [4]MRC/BHF Cardiovascular Epidemiology Unit, Department of Public Health and Primary Care, University of Cambridge, Cambridge, UK. [5]Renal Division, Department of Medicine, Faculty of Medicine and Medical Center - University of Freiburg, Freiburg, Germany. [6]Division of Quantitative Sciences, Department of Obstetrics & Gynecology, Vanderbilt Genetics Institute, Vanderbilt Epidemiology Center, Institute for Medicine and Public Health, Vanderbilt University Medical Center, Nashville, TN, USA. [7]Biomedical Laboratory Research and Development, Tennessee Valley Healthcare System (626)/Vanderbilt University, Nashville, TN, USA. [8]Target Sciences - Genetics, GlaxoSmithKline, Collegeville, PA, USA. [9]Renal Electrolyte and Hypertension Division, Department of Medicine, Department of Genetics, Perelman School of Medicine, University of Pennsylvania, Pennsylvania, PA, USA. [10]Institute for Medical Informatics, Statistics and Epidemiology, University of Leipzig, Leipzig, Germany. [11]LIFE Research Centre for Civilization Diseases, University of Leipzig, Leipzig, Germany. [12]Department of Epidemiology, Johns Hopkins Bloomberg School of Public Health, Baltimore, MD, USA. [13]Epidemiology and Clinical Research, Welch Centre for Prevention, Baltimore, MD, USA. [14]Genetics, Merck & Co., Inc., Kenilworth, NJ, USA. [15]Division of Nephrology, University of Washington, Seattle, WA, USA. [16]Kidney Research Institute, University of Washington, Seattle, WA, USA. [17]Division of Statistical Genomics, Department of Genetics, Washington University School of Medicine, St. Louis, MO, USA. [18]Saw Swee Hock School of Public Health, National University of Singapore and National University Health System, Singapore, Singapore. [19]Institute for Maternal and Child Health - IRCCS "Burlo Garofolo", Trieste, Italy. [20]Eurac Research, Institute for Biomedicine (affiliated to the University of Lübeck), Bolzano, Italy. [21]Department of Nephrology, University Hospital Regensburg, Regensburg, Germany. [22]Department of Genetic Epidemiology, University of Regensburg, Regensburg, Germany. [23]Department of Medicine, Division of Nephrology and Hypertension, University of Utah, Salt Lake City, UT, USA. [24]Medical Research Council Human Genetics Unit, Institute of Genetics and Molecular Medicine, University of Edinburgh, Edinburgh, UK. [25]Institute of Genetics and Biophysics "Adriano Buzzati-Traverso" - CNR, Naples, Italy. [26]Department of Epidemiology, Erasmus MC, University Medical Center Rotterdam, Rotterdam, The Netherlands. [27]deCODE Genetics, Amgen Inc., Reykjavik, Iceland. [28]Department of Public Health Sciences, Loyola University Chicago, Maywood, IL, USA. [29]Department of Epidemiology, University of Groningen, University Medical Center Groningen, Groningen, The Netherlands. [30]Department of Biostatistics, Johns Hopkins Bloomberg School of Public Health, Baltimore, MD, USA. [31]Department of Neurobiology, Care Sciences and Society, Division of Family Medicine and Primary Care, Karolinska Institutet, Stockholm, Sweden. [32]School of Health and Social Studies, Dalarna University, Falun, Sweden. [33]Department of Internal Medicine, Division of Nephrology, University of Groningen, University Medical Center Groningen, Groningen, The Netherlands. [34]Cardiology, Geneva University Hospitals, Geneva, Switzerland. [35]Cardiovascular Health Research Unit, Department of Medicine, University of Washington, Seattle, WA, USA. [36]Department of Biostatistics, University of Washington, Seattle, WA, USA. [37]Human Genetics Centre, University of Texas Health Science Centre, Houston, TX, USA. [38]Division of Clinical Epidemiology and Aging Research, German Cancer Research Centre (DKFZ), Heidelberg, Germany. [39]Network Aging Research, University of Heidelberg, Heidelberg, Germany. [40]Institute of Laboratory Medicine, Clinical Chemistry and Molecular Diagnostics, University of Leipzig, Leipzig, Germany. [41]Institute of Clinical Chemistry and Laboratory Medicine, University Hospital Regensburg, Regensburg, Germany. [42]Department of Biomedical Informatics, Vanderbilt University Medical Center, Nashville, TN, USA. [43]Singapore Eye Research Institute, Singapore National Eye Centre, Singapore, Singapore. [44]Ophthalmology & Visual Sciences Academic Clinical Program (Eye ACP), Duke-NUS Medical School, Singapore, Singapore. [45]Department of Ophthalmology, Yong Loo Lin School of Medicine, National University of Singapore and National University Health System, Singapore, Singapore. [46]Department of Biostatistics, University of Liverpool, Liverpool, UK. [47]Center for Primary Care and Public Health (Unisanté), University of Lausanne, Lausanne, Switzerland. [48]Department of Computational Biology, University of Lausanne, Lausanne, Switzerland. [49]Swiss Institute of Bioinformatics, Lausanne, Switzerland. [50]Department of Public Health and Primary Care, School of Clinical Medicine, University of Cambridge, Cambridge, UK. [51]Department of Clinical Epidemiology, Leiden University Medical Centre, Leiden, The Netherlands. [52]Section of Nephrology, Department of Internal Medicine, Leiden University Medical Centre, Leiden, The Netherlands. [53]Institute of Clinical Molecular Biology, Christian-Albrechts-University of Kiel, Kiel, Germany. [54]Department of Women and Child Health, Hospital for Children and Adolescents, University of Leipzig, Leipzig, Germany. [55]Centre for Pediatric Research, University of Leipzig, Leipzig, Germany. [56]Department of Biostatistics and Data Science, Wake Forest School of Medicine, Winston-Salem, NC, USA. [57]Intensive Care Medicine, Charité, Berlin, Germany. [58]Department of Nephrology and Hypertension, Friedrich-Alexander-University Erlangen-Nürnberg (FAU), Erlangen, Germany. [59]Department of Anatomy and Cell Biology, University Medicine Greifswald, Greifswald, Germany. [60]The Generation R Study Group, Erasmus MC, University Medical Center Rotterdam, Rotterdam, The Netherlands. [61]Department of Pediatrics, Erasmus MC, University Medical Center Rotterdam, Rotterdam, The Netherlands. [62]Institute of Social and Preventive Medicine (ISPM), University of Bern, Bern, Switzerland. [63]Internal Medicine - Section on Nephrology, Wake Forest School of Medicine, Winston-Salem, NC, USA. [64]Institute of Medical Informatics and Statistics, Kiel University, University Hospital Schleswig-Holstein, Kiel, USA. [65]Department of Public Health and Caring Sciences, Molecular Geriatrics, Uppsala University, Uppsala, Sweden. [66]Icelandic Heart Association, Kopavogur, Iceland. [67]Faculty of Medicine, School of Health Sciences, University of Iceland, Reykjavik, Iceland. [68]Montreal University Hospital Research Centre, CHUM, Montreal, QC, Canada. [69]Medpharmgene, Montreal, QC, Canada. [70]Laboratory of Epidemiology and Population Sciences, National Institute on Aging, Intramural Research Program, National Institutes of Health, Bethesda, MD, USA. [71]NHLBI's Framingham Heart Study, Framingham, MA, USA. [72]The Centre for Population Studies, NHLBI, Framingham, MA, USA. [73]Department of Medicine, Division of Cardiovascular Medicine, Stanford University School of Medicine, Stanford, CA, USA. [74]Stanford Cardiovascular Institute, Stanford University, Stanford, CA, USA. [75]Molecular Epidemiology and Science for Life Laboratory, Department of Medical Sciences, Uppsala University, Uppsala, Sweden. [76]Stanford Diabetes Research Center, Stanford University, Stanford, CA, USA. [77]Icelandic Heart Association, Holtasmari 1, Kopavogur IS-201, Iceland. [78]The Centre of Public Health Sciences, University of Iceland, Sturlugata 8, Reykjavík IS-101, Iceland. [79]Geisinger Research, Biomedical and Translational Informatics Institute, Rockville, MD, USA. [80]Department of Clinical Physiology, Tampere University Hospital, Tampere, Finland. [81]Department of Clinical Physiology, Finnish Cardiovascular Research Center - Tampere, Faculty of Medicine and Health Technology, Tampere University, Tampere, Finland. [82]Genome Institute of Singapore, Agency for Science Technology and Research, Singapore, Singapore. [83]Deutsches Herzzentrum München, Technische Universität München, Munich, Germany. [84]DZHK (German Centre for Cardiovascular Research), Partner Site Munich Heart Alliance, Munich, Germany. [85]Institute of Epidemiology and Biostatistics, University of Ulm, Ulm, Germany. [86]Integrated Research and Treatment Centre Adiposity Diseases, University of Leipzig, Leipzig, Germany. [87]Division of Nephrology and Hypertension, Loyola University Chicago, Chicago, IL, USA. [88]5th Department of Medicine (Nephrology, Hypertensiology, Rheumatology, Endocrinology, Diabetology), Medical Faculty Mannheim, University of Heidelberg, Mannheim, Germany. [89]Division of Genetic Epidemiology, Department of Medical Genetics, Molecular and Clinical Pharmacology, Medical University of Innsbruck, Innsbruck, Austria.

[90]Division of Biomedical Informatics and Personalized Medicine, School of Medicine, University of Colorado Denver - Anschutz Medical Campus, Aurora, CO, USA. [91]Department of Clinical Chemistry, Fimlab Laboratories, Tampere, Finland. [92]Department of Clinical Chemistry, Finnish Cardiovascular Research Center - Tampere, Faculty of Medicine and Health Technology, Tampere University, Tampere, Finland. [93]Institute of Epidemiology and Biobank Popgen, Kiel University, Kiel, Germany. [94]Diabetes Centre, Khoo Teck Puat Hospital, Singapore, Singapore. [95]Cardiovascular Epidemiology, Department of Medical Sciences, Uppsala University, Uppsala, Sweden. [96]Nuffield Department of Medicine, University of Oxford, Oxford, UK. [97]Broad Institute of Harvard and MIT, Cambridge, MA, USA. [98]Department of Medicine, Yong Loo Lin School of Medicine, National University of Singapore and National University Health System, Singapore, Singapore. [99]Wellcome Trust Centre for Human Genetics, University of Oxford, Oxford, UK. [100]Oxford Centre for Diabetes, Endocrinology and Metabolism, University of Oxford, Oxford, UK. [101]MRL, Merck & Co., Inc., Kenilworth, NJ, USA. [102]Vanderbilt University School of Medicine, Nashville, TN, USA. [103]Independent Research Group Clinical Epidemiology, Helmholtz Zentrum München, German Research Centre for Environmental Health, Neuherberg, Germany. [104]Chair of Epidemiology Ludwig- Maximilians-Universität München at UNIKA-T Augsburg, Augsburg, Germany. [105]Institute of Human Genetics, Helmholtz Zentrum München, Neuherberg, Germany. [106]Institute of Human Genetics, Technische Universität München, Munich, Germany. [107]Department of Public Health and Primary Care, Leiden University Medical Centre, Leiden, The Netherlands. [108]Institute of Genetic Epidemiology, Helmholtz Zentrum München - German Research Centre for Environmental Health, Neuherberg, Germany. [109]Chair of Genetic Epidemiology, IBE, Faculty of Medicine, LMU, Munich, Germany. [110]Department of Internal Medicine I (Cardiology), Hospital of the Ludwig-Maximilians-University (LMU) Munich, Munich, Germany. [111]Centre for Public Health Genomics, University of Virginia, Charlottesville, VA, USA. [112]Institute of Clinical Chemistry and Laboratory Medicine, University Medicine Greifswald, Greifswald, Germany. [113]Department of Cardiology, Heart Center, Tampere University Hospital, Tampere, Finland. [114]Department of Cardiology, Finnish Cardiovascular Research Center - Tampere, Faculty of Medicine and Health Technology, Tampere University, Tampere, Finland. [115]Department of Biostatistics, Boston University School of Public Health, Boston, MA, USA. [116]Section of Gerontology and Geriatrics, Department of Internal Medicine, Leiden University Medical Centre, Leiden, The Netherlands. [117]University of Maryland School of Medicine, Baltimore, MD, USA. [118]Department of Clinical Biochemistry, Landspitali University Hospital, Reykjavik, Iceland. [119]Biochemistry, Wake Forest School of Medicine, Winston-Salem, NC, USA. [120]Institute of Epidemiology, Helmholtz Zentrum München - German Research Centre for Environmental Health, Neuherberg, Germany. [121]German Center for Diabetes Research (DZD), Neuherberg, Germany. [122]Service de Néphrologie, Geneva University Hospitals, Geneva, Switzerland. [123]Einthoven Laboratory of Experimental Vascular Research, Leiden University Medical Centre, Leiden, The Netherlands. [124]Department of Genetics, University of North Carolina, Chapel Hill, NC, USA. [125]Institute of Physiology, University Medicine Greifswald, Karlsburg, Germany. [126]Department of Internal Medicine, Erasmus MC, University Medical Center Rotterdam, Rotterdam, The Netherlands. [127]Division of Endocrinology, Diabetes and Nutrition, University of Maryland School of Medicine, Baltimore, MD, USA. [128]Department of Endocrinology and Nephrology, University of Leipzig, Leipzig, Germany. [129]Duke-NUS Medical School, Singapore, Singapore. [130]The Institute for Translational Genomics and Population Sciences, Department of Pediatrics, Los Angeles Biomedical Research Institute at Harbor-UCLA Medical Center, Torrance, CA, USA. [131]Heart Centre Leipzig, Leipzig, Germany. [132]Division of Molecular Genetic Epidemiology, German Cancer Research Centre (DKFZ), Heidelberg, Germany. [133]CRCHUM, Montreal, QC, Canada. [134]Department of Cardiology, University of Groningen, University Medical Center Groningen, Groningen, The Netherlands. [135]Department of Genetics, University of Groningen, University Medical Center Groningen, Groningen, The Netherlands. [136]Durrer Centre for Cardiovascular Research, The Netherlands Heart Institute, Utrecht, The Netherlands. [137]Interfaculty Institute for Genetics and Functional Genomics, University Medicine Greifswald, Greifswald, Germany. [138]Research Unit of Molecular Epidemiology, Helmholtz Zentrum München - German Research Centre for Environmental Health, Neuherberg, Germany. [139]School of Public Health, Tongji Medical College, Huazhong University of Science and Technology, Wuhan, China. [140]Vanderbilt University Medical Centre, Division of Nephrology & Hypertension, Nashville, TN, USA. [141]Department of Ophthalmology, Tohoku University Graduate School of Medicine, Sendai, Japan. [142]Anatomic Pathology, University of Washington Medical Center, Seattle, WA, USA. [143]Department of Nephrology, Diabetology and Rheumatology, Kliniken Südostbayern, Traunstein, Germany. [144]Institute of Physiology, University of Zurich, Zurich, Switzerland. [145]Division of Epidemiology, Department of Medicine, Vanderbilt Genetics Institute, Vanderbilt University Medical Centre, Nashville, TN, USA. [146]Department of Veteran's Affairs, Tennessee Valley Healthcare System (626)/Vanderbilt University, Nashville, TN, USA. [147]Kidney Health Research Institute (KHRI), Geisinger, Danville, PA, USA. [148]Department of Nephrology, Geisinger, Danville, PA, USA. [149]Division of Kidney, Urologic and Hematologic Diseases, National Institute of Diabetes and Digestive and Kidney Diseases, National Institutes of Health, Bethesda, MD, USA. [150]Department of Medicine, University of Maryland School of Medicine, Baltimore, MD, USA. [151]Geisinger Research, Biomedical and Translational Informatics Institute, Danville, PA, USA. [152]Cardiovascular Health Research Unit, Department of Medicine, Department of Epidemiology, Department of Health Service, University of Washington, Seattle, WA, USA. [153]Kaiser Permanente Washington Health Research Institute, Seattle, WA, USA. [154]Department of Pediatrics, Harbor-UCLA Medical Centre, Torrance, CA, USA. [155]Department of Medicine, Harbor-UCLA Medical Centre, Torrance, CA, USA. [156]Department of Physiology and Biophysics, University of Mississippi Medical Centre, Jackson, MS, USA. [157]National Institute for Health Research Blood and Transplant Research Unit in Donor Health and Genomics, University of Cambridge, Cambridge, UK. [158]Present address: Celgene Inc., Cambridge, MA, USA. [159]Present address: Biogen Inc., Cambridge, MA, USA. [160]These authors contributed equally: Alexander Teumer, Yong Li, Sahar Ghasemi, Bram P. Prins, Matthias Wuttke, Tobias Hermle. [161]These authors jointly supervised: Adam S. Butterworth, Adriana M. Hung, Cristian Pattaro, Anna Köttgen.

