## [Peer Review File · Nature Communications]

Reviewers' Comments:

Reviewer #1:

Remarks to the Author:

These investigators present results of a trans-ethnic GWA metaanalysis (GWAMA) on 564,257 subjects that identified 68 urine albumin creatinine ratio (UACR)-associated loci. Additional analyses evaluated the association of these loci with the categorical microalbuminuria phenotype, assessed ancestry-specific loci in European ancestry subjects and African-Americans, and identified diabetic-specific association loci. The UACR loci were then annotated using bioinformatics tools for functional enrichment, statistically fine mapped and colocalized with known tissue eQTLs to define critical intervals and driver genes. A PheWas was performed on EA subjects in the Million Veterans Program, which identified traits that associated with UACR loci. *Drosophila* nephrocytes were used to characterize function sequelae of knock-down of 3 candidate UACR-associated genes. Pathological albuminuria (thresholds to be defined) associates with deleterious cardiovascular (CVD) and kidney disease phenotypes as well as CVD and all cause mortality. The mechanistic links responsible for these association are poorly understood. Given that, this study is a tour de force. Its subject size and characterization of UACR-associated loci provide an in depth and robust picture of the genetic architecture underlying this phenotype and its correlated traits. In addition, the data roadmap hypothetical functions for genes, poorly characterized experimentally to date, which can now be experimentally test. These data will be a resource for the investigative community interested in albuminuria. Given the track record of this consortium, the experimental design and analysis are rigorous. However, I have the following specific comments:

1. The association of albuminuria with morbid phenotypes, perhaps with the exception of kidney diseases, remains a conundrum. Why albuminuria is so robustly associated with CVD and CVD- and all-cause mortality remains a conundrum. After reading the manuscript, putative mechanisms remain obscure. The reader is confronted with a deluge of data and association lists that may provide novel insights into disease but not an integrated hypothesis for mechanisms driving the association with traits. I still am left with the question of whether albuminuria is a convenient biomarker or a clue to a causal pathway(s).
2. These data really focus on median UACRs that are normal in most subjects. In fact, the 25th and 75th percentiles suggest very few subjects had UACRs > 100, placing the subjects in a UACR range where even those individuals with microalbuminuria might reverted to normal [extrapolating from longitudinal studies in diabetic subjects]. Conceivably many of loci, particularly the highly significant loci like CUBN, might map mechanisms of normal albumin handling but provide little insight into the deleterious phenotypes that the authors reference in this manuscript. I note that novel UACR loci were identified in diabetic subjects (ST6) and thought the diabetic-specific loci might point to these pathological mechanisms. However even the diabetic subjects had low UACRs (ST1), although a number would likely develop heavier proteinuria with time.
3. The label "transethnic" is a bit of a reach. The subjects in these analyses are overwhelmingly of European ancestry. The authors should at least indicate if the UACR-associated loci directionally concordant effects in the subjects of non-European ancestry, although I recognize statistically significant ancestry-related heterogeneity was not identified.
4. The results presenting the lack of association of APOL1 kidney risk variants and UACR is a bit misleading. The text states an association was not found and notes this likely reflects the additive model used by GWAS. This could mislead readers not expert in these types of analyses; the APOL1 association with UACR and/or microalbuminuria has been repeatedly shown under a recessive model of inheritance (I suspect in many of the AA cohorts in this study) and not using an additive model. Even the investigators want to comment on APOL1, they should assess the association using a recessive model.
5. The *Drosophila* studies are a bit inconsistent. The atypical PKC, PRKC1, disrupts slit diaphragm architecture (Figure 8), a barrier to albumin loss and a podocyte phenotype, but it also reduced albumin uptake, a tubular epithelial cell phenotype (supplemental Figure 8). Is PRCK1, expressed in tubular epithelia?
6. Text in the Introduction (lines 336-337) and Discussion (lines 703 to 707) could be interpreted that reducing albuminuria is causally related to reduction of kidney disease or cardiovascular

disease. I believe these observations remain associations with little evidence that albuminuria reduction is causally linked to risk reduction. Experimental evidence shows the drug classes used in this studies impact heart and kidney tissue injury, e.g., by reducing fibrogenesis.

7. I concur with the investigators that these data will be a valuable resource for the investigative community studying albuminuria and its associations with disease traits. I would encourage the Consortium to make more of their datasets as well as their code freely available. These analyses cannot be replicated with independent datasets, but outside investigators should have the opportunity to replicate the findings as well as determine if additional insights can be discovered. Making the data and code accessible in a format to facilitate these goals is desirable.

Minor comments:

1. Page 14, line 494: "lipoid" should be lipid
2. Supplementary Figure 8, panel d': "surface" should be surface
3. Include closest gene names supplemental table 4.
4. Supplemental Table 1: Define columns P and Q. What is the difference between "Sample size of MA cases" and "Total sample size MA?"

Reviewer #2:

Remarks to the Author:

This is a very interesting and comprehensive article. The GWAS is suitably powered, with wide (though not perfect) ethnic coverage. Reassurance is given statistically to show that the results are not overly influenced by UKBioBank predominance in the sampling. It is interesting that some loci mapped to creatine, some to albumin, and only two to both. The authors reassuringly found the previously known loci, together with multiple further loci, many of which are highly plausible.

Q: do the authors believe this is a saturation screen? Will it be necessary to repeat it in the future, or has this found the answers we need?

I liked the way this study linked into and leveraged data from, other studies like INTERVAL

Line 337: disease should be disease

It would be nice to name each gene that's the topic of extensive discussion, in full on the first use – as well as in Table 1.

What is the basis for gene inclusion in Table 1? Are these all genes encompassing the most informative SNP? Or the genes in the centre of the lod peak? Are they proven to be involved, or merely the authors' selection of likeliest candidate genes within the interval? This should be stated.

The Drosophila work is interesting, but under-described and under-interpreted, and feels like a bolt-on at present.

Where are the knockdown values for the RNAi lines chosen? The data cannot be used without good qPCR validation. Unless this is better than 70% knockdown in garland cells, I would not trust the data. Is evidence for protein knockdown available? That would provide valuable support. More details of crosses, temperatures and rearing are needed.

figure 8: The lack of effect of Drosophila verprolin 1 (Vrp1) knockdown on the Drosophila Verprolin is not surprising to this reviewer. Any similarity between human WIPF3 and Vrp1 proteins is really low grade, and most orthology resources do not call the pairing – WIPF1 fares (slightly) better (see FLYBase orthology summary So homology certainly cannot be asserted, even if someone published before. Rather, the lack of homology could be used to explain this negative result.

The authors have done a good job of weaving through the logic of the insect nephrocytes; renal

they are not, but they replicate two separate processes (membrane specialisation and endocytosis) that are present in the kidney, and assigning oaf to one class, and aPKC to the other. It would be nice to include a phrase 'Although albumin is not a natural element of Drosophila body fluids, labelled mammalian albumin is taken up by nephrocytes, forming a useful assay...'

It would be nice for the authors to discuss in a little more detail how they think oaf and aPKC both inhibit albumin uptake. If the permeability barrier is disrupted, for example, shouldn't that make it easier for albumin to reach the internalization site? Or is the point of the slit diaphragm in nephrocytes simply to set up a specialised cytoskeletal region in which endocytosis can be performed easily?

Reviewer #3:

Remarks to the Author:

Teumer et al. report a trans-ethnic genome-wide meta-analysis of urinary albumin-to-creatinine ratio (UACR; a biomarker of renal and cardiovascular disease) in ~560,000 people (50% larger than the largest study to date; Haas et al. AJHG 2018), including ~50,000 with diabetes. A comprehensive array of follow-up analyses included electronic health record associations in an independent dataset, fine-mapping, multi-omics integration, and in vitro work. They identify a total of 68 associated loci (more than double the number in the largest analysis to date) and a number of candidate causal genes for these associations.

This is a high-quality study and a well-written paper. A few comments and suggestions for further improvement are below.

1-Inclusion/exclusion criteria for participants: From the Methods section, it is not immediately clear if people with clinically-defined microalbuminuria (MA) or those with diabetes mellitus (DM) have been included in the main meta-analysis or whether there were other inclusion/exclusion criteria. This should be noted at the beginning of the methods section in the part that describes the cohorts.

2-Directionality of associations: It would be helpful if the authors could provide some additional analyses and insights on whether the genetic associations they observe are cause or consequence of major diseases outcomes related to kidney dysfunction (CKD, coronary disease, stroke, hypertension, heart failure), i.e. whether directionality of association is (a) gene variant -> impaired kidney function -> disease or (b) gene variant -> disease (e.g. DM) -> impaired kidney function. This could be achieved with a combination of: (1) Mendelian randomization analyses of polygenic score for UACR -> outcomes (using UK Biobank or summary statistics from previous GWAS); (2) Mendelian randomization analyses of disease polygenic scores to UACR; (3) analyses of effect size on UACR (in people without disease, which could be done in UKBB at least) vs effect size on outcome (from UKBB or GWAS summary statistics); (4) identification of variants that appear to have a primary effect on disease risk and secondary on UACR vs those that appear to have a primary effect on UACR. This may also help interpret some of the potential for therapeutic translation of the GWAS findings.

3-Genetic association across the spectrum of UACR variation. It'd be interesting to know if the identified variants are associated with UACR homogeneously across the phenotypic variation of UACR in the population or if effect sizes tend to be stronger (or weaker) in people with extreme UACR. This can be done using conditional quantile regression and meta-regression as done by Abadi et al. (Am J Hum Genet. 2017; 101:925-938) using individual level genotype data and UACR data from UK Biobank. Another complementary approach could be to divide the UKBB population in quantiles of the polygenic score and study the association with clinically-defined microalbuminuria for different quantiles relative to the bottom one (low-genetic risk). Both analyses could be quite insightful.

Minor

-It has been disputed (see for instance Nat Genet. 2016;48:314-7) whether the conventional threshold for genome-wide significant 5×10^{-8} holds with more densely imputed panels (as in this study). Wonder if authors could report in a supplementary table how many loci would survive one of the more stringent thresholds that have been proposed or whether any additional variants would meet an annotation-specific threshold (lower for protein truncating variants or missense variants as suggested in the Nat Genet. 2016;48:314-7 paper).

-It'd be interesting to know how much of the (low) heritability of UACR is explained by the hits reported here. Please, calculate and report heritability explained.

-Fine mapping: the preliminary step of the fine-mapping uses approximate conditional analyses with GCTA, which is a reasonable first-line approach but is prone to error particularly for lower frequency variants where the pattern of LD is less precisely measured by reference panels and may be imperfectly captured by the r -squared. Some of the conditionally-independent variants identified by GCTA at the CUBN locus are about 1% allele frequency. At this locus and at the other ones where more than one signal are suggested by GCTA, the results of approximate conditional analyses should ideally be confirmed with formal conditional analyses (eg. using individual-level genotypes in UKBB). At the bare minimum, authors could show that the conditionally-independent variants put forward by GCTA are actually conditionally-independent of each other using individual-level data (a more relaxed p -value threshold can be used to account for the loss of sample size).

-The abstract and discussion report 68 loci. After reading the main text, it seems that these stem from 59 trans-ethnic meta-analysis loci + 5 additional from ethnic specific analysis + 4 additional from diabetes-specific analysis. It would be helpful if this was explicitly stated in the main text at the beginning of the results and depicted in Figure S1.

-In Figure S1, please add a panel C with the workflow of the diabetes-specific analysis. A separate additional panel D could be added with a Venn diagram showing the overlap of the identified loci in the three strands of the study.

-Line 441: for binomial test, please provide number of variants that were found to have a concordant direction.

-Line 412: $p < 0.05$ and same direction? If some are in different direction, please report only the number of variants with directional concordance.

-Line 416: "mostly population based" should read "other cohorts, most of which are population-based". Correct?

-LDSC-regression intercept values should be used throughout in lieu of lambda GC, which does not distinguish genetic confounding from signal. Given the size of the study, attenuation ratio and mean chi-squared statistics should be reported in addition to LDSC-regression intercept values (as recommended in Nat Genet. 2018 Jul;50(7):906-908).

-Line 424: "suggestive" should read "association that did not meet the genome-wide significance threshold".

-Line 425: please provide p for each SNP. Could you test recessive inheritance or association in heterozygous or homozygous variant carriers versus homozygous non-carriers in UKBB?

-Line 494: "lipid" is misspelled.

Dear Reviewers,

We thank you for the thorough evaluation of our manuscript, and for the constructive and helpful comments. We have carefully considered them, and respond to each comment in the point-by-point response, below. We believe that your comments have helped us to significantly improve our manuscript, and hope that the implemented changes meet your requests and that our manuscript may now be acceptable for publication.

Sincerely,

Alexander Teumer, Cristian Pattaro, and Anna Köttgen, on behalf of all authors

Point-by-point response

Reviewer #1:

These investigators present results of a trans-ethnic GWA metaanalysis (GWAMA) on 564,257 subjects that identified 68 urine albumin creatinine ratio (UACR)-associated loci. Additional analyses evaluated the association of these loci with the categorical microalbuminuria phenotype, assessed ancestry-specific loci in European ancestry subjects and African-Americans, and identified diabetic-specific association loci. The UACR loci were then annotated using bioinformatics tools for functional enrichment, statistically fine mapped and colocalized with known tissue eQTLs to define critical intervals and driver genes. A PheWas was performed on EA subjects in the Million Veterans Program, which identified traits that associated with UACR loci. *Drosophila* nephrocytes were used to characterized function sequelae of knock-down of 3 candidate UACR-associated genes.

Pathological albuminuria (thresholds to be defined) associates with deleterious cardiovascular (CVD) and kidney disease phenotypes as well as CVD and all cause mortality. The mechanistic links responsible for these association are poorly understood. Given that, this study is a tour de force. Its subject size and characterization of UACR-associated loci provide an in depth and robust picture of the genetic architecture underlying this phenotype and its correlated traits. In addition, the data roadmap hypothetical functions for genes, poorly characterized experimentally to date, which can now be experimentally test. These data will be a resource for the investigative community interested in albuminuria. Given the track record of this consortium, the experimental design and analysis are rigorous.

Response: We thank the Reviewer for the appreciation of our work.

However, I have the following specific comments:

1. The association of albuminuria with morbid phenotypes, perhaps with the exception of kidney diseases, remains a conundrum. Why albuminuria is so robustly associated with CVD and CVD- and all-cause mortality remains a conundrum. After reading the manuscript, putative mechanisms remain obscure. The reader is confronted with a deluge of data and association lists that may provide novel insights into disease but not an integrated hypothesis for mechanisms driving the association with traits. I still am left with the question of whether albuminuria is a convenient biomarker or a clue to a causal pathway(s).

Response: We agree with the Reviewer that the illumination of mechanisms linking albuminuria to cardiovascular morbidity and mortality is an important endeavor. Based on the data presented in our manuscript as well as on published literature¹, we believe that not all of the UACR-associated genes operate in the same pathway to influence UACR. It may therefore be challenging to identify a single

integrated hypothesis why albuminuria is associated with increased CVD risk. For example, we know from experimental studies that mutations in *COL4A4*, a locus identified in our study, are linked to albuminuria through an altered composition of the glomerular basement membrane², whereas genetic variants in *CUBN* are linked to albuminuria via altered tubular reabsorption of filtered albumin. In addition, there may be genetic variants such as those in *GATM* that are associated with UACR because they are linked to urinary creatinine rather than albumin concentrations, thereby influencing UACR.

We used diverse approaches to provide readers with information about potential mechanism connecting each of the identified genes to UACR: first, we performed co-localization analyses with gene expression separately in human glomerular and tubular kidney portions. This shows, for example, that the UACR association signal co-localizes with *PRKCI* expression in glomerular portions, while it co-localizes with *MUC1* expression in tubular kidney portions. This is consistent with existing knowledge about their localization and pathophysiology, and illustrates how co-localization experiments can connect genes to relevant tissues. Second, we performed annotation of fine-mapped UACR-associated SNPs using human kidney cell-type specific DNA accessibility maps, which can reveal cell-type specific regulatory variants. Third, we evaluated the association of each UACR-associated index SNP separately with urinary creatinine and albumin, which can highlight those UACR-associations primarily driven through an association with urinary albumin (Figure 2C). Lastly, we experimentally examined several candidates from our screen using the *Drosophila* nephrocyte model to assess whether evolutionarily conserved UACR-associated genes operate through filtration or reabsorption-like mechanisms. Together, this characterization contains a lot of useful information about potential mechanisms driving the association with UACR at each locus, which we have now discussed in more detail (page 21, line 802ff). More detailed experimental characterizations of the genes underlying the identified association signals are required to understand underlying molecular mechanisms in detail, and represent multi-year follow-up projects.

While the mechanisms connecting the genes to UACR may differ, it is still possible that the association of UACR with CVD operates mainly through one integrated mechanism. To evaluate this question, we had performed gene set enrichment analyses, summarized in Supplementary Table 8 (Supplementary Table 8). Among the significantly enriched gene sets (FDR <0.05), there were many related to abnormal vascular morphology including during development, to heart development and morphology, and to *in utero* development, which is highly dependent on the vasculature of the placenta. The connection of UACR to the endothelium and blood vessels is further supported by the results from the PheWAS (Supplementary Table 9) and the genetic correlation analyses (Supplementary Table 10), which highlight a connection of UACR-associated genetic variants to (essential) hypertension. Very recently, Haas and colleagues published a Mendelian Randomization analysis¹ and reported that elevated UACR was causally associated with increased risk of hypertension, and vice versa that elevated blood pressure causes more albuminuria. This is consistent with a feed-forward loop between albuminuria and blood pressure and suggests that albuminuria could increase risk of cardiovascular disease through blood pressure. Of note, according to Supplementary Figure 2 of Haas *et al*, SNPs in *CUBN* and *CYP1A1* showed strong association with UACR but not blood pressure. These SNPs are related to UACR via the tubular reabsorption of albumin and an association with urinary creatinine, respectively, in our data. This may indicate that the increased filtration of albumin in the glomerulus, potentially as a result of endothelial damage, and not albuminuria *per se* may represent the mechanism that links albuminuria to hypertension and increased CVD risk. We have now added these observations to the discussion on page 21, line 808ff.

2. These data really focus on median UACRs that are normal in most subjects. In fact, the 25th and 75th percentiles suggest very few subjects had UACRs > 100, placing the subjects in a UACR range where even those individuals with microalbuminuria might revert to normal [extrapolating from

longitudinal studies in diabetic subjects]. Conceivably many of loci, particularly the highly significant loci like CUBN, might map mechanisms of normal albumin handling but provide little insight into the deleterious phenotypes that the authors reference in this manuscript. I note that novel UACR loci were identified in diabetic subjects (Supplementary Table 6) and thought the diabetic-specific loci might point to these pathological mechanisms. However even the diabetic subjects had low UACRs (Supplementary Table 1), although a number would likely develop heavier proteinuria with time.

Response: We agree that it is important to point out that our meta-analysis is based on mostly population-based studies with median UACR in the normal range (UACR <30 mg/g). We do observe, however, that most of the SNPs discovered in association with UACR also showed an association with microalbuminuria (UACR >30 mg/g, Figure 2B). The number of individuals with macroalbuminuria / nephrotic range proteinuria is too small in population-based cohorts to be studied separately. We have therefore included a statement in the discussion that it is presently unclear whether these loci translate to nephrotic range proteinuria, which should be the focus of complementary studies (page 22, line 828). Another line of evidence that could be followed separately is to study the effect of full knockouts of the discovered genes in experimental models. Such a full knockout may have a more dramatic effect on albuminuria than the associated SNPs identified in our study, which may have risen to high population frequencies because they represent hypomorphic alleles with more modest effects on the phenotype.

3. The label “transethnic” is a bit of a reach. The subjects in these analyses are overwhelmingly of European ancestry. The authors should at least indicate if the UACR-associated loci directionally concordant effects in the subjects of non-European ancestry, although I recognize statistically significant ancestry-related heterogeneity was not identified.

Response: As suggested, we have now included an additional column in Supplementary Table 3 (“Anc direction”), in which we indicate the direction of association in the individual ancestry groups. In addition, we have removed “trans-ethnic” from the manuscript title.

4. The results presenting the lack of association of APOL1 kidney risk variants and UACR is a bit misleading. The text states an association was not found and notes this likely reflects the additive model used by GWAS. This could mislead readers not expert in these types of analyses; the APOL1 association with UACR and/or microalbuminuria has been repeatedly shown under a recessive model of inheritance (I suspect in many of the AA cohorts in this study) and not using an additive model. Even the investigators want to comment on APOL1, they should assess the association using a recessive model.

Response: We agree with the Reviewer that these results could potentially be misleading. Given the small African American sample, the additive model commonly used for discovery screens in GWAS, as well as the use of imputed rather than directly genotyped information for these low-frequency variants, we have decided to remove the mention of *APOL1* from the manuscript and changed the manuscript accordingly (page 12, line 445ff).

For the Reviewer’s information, we have investigated potential reasons why the known *APOL1* CKD risk alleles do not achieve genome-wide significance in our meta-analysis. Based on sensitivity analyses among 1,605 AA participants of the ARIC study with both imputed and directly genotyped *APOL1* variants, we found that the association p-value of an additive genetic model was similar to the one obtained from a recessive model, although no conclusive answer could be given due to the low frequency of the risk variants. Many previous studies directly genotyped the *APOL1* risk variants. In the ARIC Study, the use of imputed compared to directly genotyped data led to smaller effect sizes

and greater standard errors, resulting in less significant associations with UACR. Lastly, our study was conducted in the general population, where UACR levels are much lower than among individuals presenting with focal-segmental glomerulosclerosis, who were examined in many previous studies.

5. The *Drosophila* studies are a bit inconsistent. The atypical PKC, PRKC1, disrupts slit diaphragm architecture (Figure 8), a barrier to albumin loss and a podocyte phenotype, but it also reduced albumin uptake, a tubular epithelial cell phenotype (supplemental Figure 8). Is PRCK1, expressed in tubular epithelia?

Response: We understand that a combination of reduced FITC-albumin endocytosis and disrupted slit diaphragm morphology upon *PRKC1*-silencing (Suppl. Fig. 8) may seem inconsistent at first, but this arises merely as a consequence of differences in anatomy and function between the mammalian kidney and the *Drosophila* nephrocyte. Proper slit diaphragm function is a prerequisite for the formation of the nephrocyte's complex network of membrane invaginations called labyrinthine channels. Endocytosis of FITC-albumin in nephrocytes occurs within these labyrinthine channels. Indirectly, probably through loss of surface area, a disturbance of the nephrocyte 3D-architecture results in a drastic decline of FITC-albumin uptake³. In nephrocytes, silencing slit diaphragm proteins will therefore cause a reduction and not an increase of FITC-albumin endocytosis. As reduced albumin uptake in this model can result either from loss of proper slit diaphragms (glomerular component) or from reduced endocytosis (tubular component), we refer to the FITC-albumin endocytosis in nephrocytes as an integrative read-out (page 18, line 693). To distinguish whether the observed reduced albumin uptake resulted from impaired slit diaphragm formation or from reduced endocytosis, we conducted immunofluorescence studies of slit diaphragm proteins, which are only expected to be altered when slit diaphragm formation is impaired. These experiments showed that knockdown of *prkci* led to impaired slit diaphragm formation, the glomerular component of the integrative readout. We have now clarified this point on page 19, lines 715).

To address the Reviewer's second question, we have queried the expression of *PRKCI* in different kidney cell types from publicly available RNA-sequencing data from both human⁴ and mouse⁵. While we found that there is low expression of *PRKCI* also in tubular cell types, gene expression in podocytes was much higher: the z-score scaled expression in podocytes/tubular cells was 3.2/-0.17 for mouse and 2.1/-0.79 for human.

6. Text in the Introduction (lines 336-337) and Discussion (lines 703 to 707) could be interpreted that reducing albuminuria is causally related to reduction of kidney disease or cardiovascular disease. I believe these observations remain associations with little evidence that albuminuria reduction is causally linked to risk reduction. Experimental evidence shows the drug classes used in this studies impact heart and kidney tissue injury, e.g., by reducing fibrogenesis.

Response: We have now rephrased the two sections to reflect the Reviewer's comment (page 9, line 334 and page 21, lines 789-793).

7. I concur with the investigators that these data will be a valuable resource for the investigative community studying albuminuria and its associations with disease traits. I would encourage the Consortium to make more of their datasets as well as their code freely available. These analyses cannot be replicated with independent datasets, but outside investigators should have the opportunity to replicate the findings as well as determine if additional insights can be discovered. Making the data and code accessible in a format to facilitate these goals is desirable.

Response: We absolutely agree with the Reviewer. Upon publication of our article, we will share full genome-wide association statistics for UACR (overall, European ancestry, and among those with diabetes) as well as for MA, through our consortium website as well as through dbGaP (study accession number phs000930.v7.p1). The GWAS summary statistics data sharing resource of the CHARGE Consortium through dbGaP has been described in a recent publication in Nature Genetics (2016; 48(7):702-3). For now, we have set up a temporary solution for data access, where the Reviewers can view the summary statistics exactly as deposited until they become available through dbGaP (please note that files have to be referenced directly using an encrypted connection):

https://transfer.sysepi.medizin.uni-greifswald.de/pr181015/formatted_20170020-UACR_DM-ALL-nstud_18-SumMac_400.tbl.rsid.gz

https://transfer.sysepi.medizin.uni-greifswald.de/pr181015/formatted_20170711-UACR_overall-ALL-nstud_27-sumMac_400.tbl.rsid.gz

https://transfer.sysepi.medizin.uni-greifswald.de/pr181015/formatted_20180205-MA_overall-ALL-nstud_18-SumMac_400.tbl.rsid.gz

https://transfer.sysepi.medizin.uni-greifswald.de/pr181015/formatted_20180517-UACR_overall-EA-nstud_18-SumMac_400.tbl.rsid.gz

For the statistical analyses, we used freely available software programs and report them along with the options in Supplementary Table 2 as well as in the methods part. Shell scripts to call on these programs and code to produce figures will be shared upon request. Lastly, we have made publicly available both our analysis plan for this project (https://ckdgen.eurac.edu/mediawiki/index.php/CKDGen_Round_4_EPACTS_analysis_plan) as well as the code used to generate the phenotypes (<https://github.com/genepi-freiburg/ckdgen-pheno>).

Minor comments:

1. Page 14, line 494: “lipoid” should be lipid

Response: The term “lipoid” metabolism is a fixed component of the phenotype codes used for phenome-wide association studies in the R package for these analyses. It is based on the ICD9 code 272 and its derivatives. The general, upper-level description for this code is “Disorders of lipoid metabolism” (see phewascatalog.org), with lipoid capturing a whole range of lipid-related (lipoid) disorders. We have described the use of the R package ‘PheWAS’ on page 27, lines 959-960, and cited the corresponding publication.

2. Supplementary Figure 8, panel d’: “surface” should be surface

Response: Thank you for bringing this to our attention, we have fixed this.

3. Include closest gene names supplemental table 4.

Response: We have added this information to Supplementary Table 4.

4. Supplemental Table 1: Define columns P and Q. What is the difference between “Sample size of MA cases” and “Total sample size MA?”

Response: “Sample size of MA cases” refers to the number of individuals with microalbuminuria (UACR >30 mg/g) in the GWAS of microalbuminuria presence vs. absence, whereas “total sample size MA” refers to the total number of analyzed individuals (those with and without microalbuminuria). We have clarified this in the header and in a footnote of the table.

Reviewer #2:

This is a very interesting and comprehensive article. The GWAS is suitably powered, with wide (though not perfect) ethnic coverage. Reassurance is given statistically to show that the results are not overly influenced by UKBioBank predominance in the sampling. It is interesting that some loci mapped to creatine, some to albumin, and only two to both. The authors reassuringly found the previously known loci, together with multiple further loci, many of which are highly plausible.

Q: do the authors believe this is a saturation screen? Will it be necessary to repeat it in the future, or has this found the answers we need?

Response: The Reviewer is raising an interesting point. We do not believe that our study constitutes a saturation screen. Over the past ten years, it has been shown that further increases in sample lead to additional discoveries of genetic loci for polygenic complex traits and diseases (compare for example Figure 2 in Ahlqvist *et al.*⁶. With better statistical power, additional loci of increasingly smaller effect size will be discovered, which may be more distantly related to central genes influencing albuminuria. This may mean that these “non-central” genes are less important for albuminuria because of the potential ability to compensate for their altered function. However, this would need to be confirmed experimentally because a small effect on the phenotype can also indicate that the gene harbors only hypomorphic genetic variants, but a full loss or gain of function could still have a substantial effect.

Reviewer Figure 1: Expected number of discoveries at increasing GWAS discovery sample sizes.

To address the Reviewer’s question quantitatively, we have performed additional analyses using a recently published method to estimate the number of expected discoveries and the corresponding percentage of GWAS heritability explained with increases in sample size using the summary statistics of our UACR meta-analysis as input⁷. As shown in **Reviewer Figure 1**, we can expect that with increasingly larger sample sizes, additional genetic loci for UACR will be detected. The corresponding publication discusses that, especially when the sample size is not very large, the method underestimates the number of discoveries when there are many SNPs of small effect, as is the case with UACR. This may explain the discrepancy between the predicted and observed number of

discoveries at our current sample size (blue vertical line). We have added this information to the manuscript on page 11, line 417 (Results), page 26, line 940 ff (Methods), and are showing the resulting plots as a new **Supplementary Figure 3**.

I liked the way this study linked into and leveraged data from, other studies like INTERVAL

Response: Thank you for the appreciation of our work.

Line 337: disease should be disease

Response: Thank you, we corrected the typo.

It would be nice to name each gene that's the topic of extensive discussion, in full on the first use – as well as in Table 1.

Response: We have amended the manuscript and Table 1 accordingly.

What is the basis for gene inclusion in Table 1? Are these all genes encompassing the most informative SNP? Or the genes in the centre of the lod peak? Are they proven to be involved, or merely the authors' selection of likeliest candidate genes within the interval? This should be stated.

Response: We have now clarified in the table header that genes were included in Table 1 when supported by missense variants with high posterior probability of association, missense variants mapping into small credible sets, or with high posterior probability of co-localization of the UACR association signal and gene expression in kidney tissues.

The *Drosophila* work is interesting, but under-described and under-interpreted, and feels like a bolt-on at present.

Response:

Thank you for bringing this to our attention, we have now substantially expanded the description of the *Drosophila* experiments in the manuscript (Results, page 18, lines 672 - page 19, line 716).

Where are the knockdown values for the RNAi lines chosen? The data cannot be used without good qPCR validation. Unless this is better than 70% knockdown in garland cells, I would not trust the data. Is evidence for protein knockdown available? That would provide valuable support.

Response: While qPCR may be used to confirm silencing on the transcript level in *Drosophila*, this is technically limited in nephrocytes. A *Drosophila* larva contains only about 25-30 individual garland cell nephrocytes within the entire organism. For extraction, these cells need to be attached to the much larger anatomical structure of the proventriculus and cannot be handled separately. It is therefore not possible to obtain sufficient material to run qPCR exclusively from nephrocytes. Observing a similar phenotype using two independent transgenic RNAi lines is commonly considered to confirm specificity. If the knockdown was insufficient, we should expect a lack of phenotypic

response. In contrast, the two non-overlapping *oaf*-RNAi lines caused a reduction of nephrocyte FITC-albumin endocytosis while not affecting slit diaphragm morphology. We believe this identical and uncommon phenotypic result using two different transgenic RNAi lines appears specific with a high level of confidence.

If antibodies directed against the *Drosophila* gene are available, staining nephrocytes directly is a good way to confirm efficient silencing on the protein level. An antibody directed against human PKC zeta (SC20, Santa Cruz) has been described to faithfully detect *Drosophila* aPKC while lacking a signal in null mutants⁸, and the PKC zeta-antibody has been repeatedly employed to detect the *Drosophila* protein⁹. We now added stainings of nephrocytes using this antibody (Supplementary Figure 9A-C, to the previous Supplementary Figure 8). We observed a strong reduction of the signal upon expression of aPKC-RNAi compared to control. This confirms an efficient knockdown on the protein level for aPKC. Unfortunately, there is no antibody available for the *Drosophila* *oaf* protein.

More details of crosses, temperatures and rearing are needed.

Response: As previously noted in the materials and methods, we employed the GAL4/UAS system using the stocks identified by their unique stock numbers from two major stock centers and flies were raised at 30°C. The GAL4/UAS-system is a binary expression system that employs the yeast promoter GAL4 to direct expression of desired transgenes succeeding the so-called upstream activating sequence (UAS). The UAS sequence does not occur naturally within the *Drosophila* genome, thus commonly neither the presence of GAL4 nor of UAS-constructs alone entails significant consequence, which allows maintaining transgenic lines that are phenotypically bland. In every single case, we crossed the transgenic GAL4-line with the respective UAS-line and the F1 generation was studied in the experiments. Harboring both transgenes, the F1 exhibits induction of the expression of the transgene that follows the UAS sequence. This reflects a standard approach; thus it is quite uncommon to include extensive details regarding such crosses within the manuscript. As noted, we employed *Dorothy-GAL4* (BDSC #6903) to induce transgene expression within nephrocytes using the indicated UAS lines in that way. We expanded the information regarding the crosses in the Methods section (page 31, line 1086ff) by noting the control RNAi (EGFP, BSDC# 41553), the food composition (standard agar cornmeal molasses) and the developmental stage (wandering third instar).

figure 8: The lack of effect of *Drosophila* verprolin 1 (*Vrp1*) knockdown on the *Drosophila* Verprolin is not surprising to this reviewer. Any similarity between human WIPF3 and *Vrp1* proteins is really low grade, and most orthology resources do not call the pairing – WIPF1 fares (slightly) better (see FLYBase orthology summary). So homology certainly cannot be asserted, even if someone published before. Rather, the lack of homology could be used to explain this negative result.

Response: We agree with the Reviewer that the orthology of *Vrp1* is uncertain, despite the literature and the shared functional domains. We adjusted the manuscript text accordingly (page 20, line 777).

The authors have done a good job of weaving through the logic of the insect nephrocytes; renal they are not, but they replicate two separate processes (membrane specialisation and endocytosis) that are present in the kidney, and assigning *oaf* to one class, and aPKC to the other. It would be nice to include a phrase 'Although albumin is not a natural element of *Drosophila* body fluids, labelled mammalian albumin is taken up by nephrocytes, forming a useful assay...'

Response: We agree with the Reviewer, and have added the phrase to the manuscript (page 18, line 684).

It would be nice for the authors to discuss in a little more detail how they think oaf and aPKC both inhibit albumin uptake. If the permeability barrier is disrupted, for example, shouldn't that make it easier for albumin to reach the internalization site? Or is the point of the slit diaphragm in nephrocytes simply to set up a specialised cytoskeletal region in which endocytosis can be performed easily?

Response: As pointed out above, FITC-albumin uptake occurs within the labyrinthine channels that in turn depend on proper slit diaphragms. Loss of functional slit diaphragms thus results in a drastic loss of surface area for the uptake, which likely reduces albumin uptake indirectly. The size cut-off for slit diaphragm passage in the nephrocyte is slightly higher, with albumin being just below the upper size-limit for passage³. Thus loss of slit diaphragms does not enable a passage that was previously precluded. We now note this information in the manuscript (page 18, line 681): "Studying endocytosis of a tracer molecule able to pass the slit diaphragm, such as albumin, in nephrocytes renders an integrative read-out of nephrocyte function".

Reviewer #3:

Teumer et al. report a trans-ethnic genome-wide meta-analysis of urinary albumin-to-creatinine ratio (UACR; a biomarker of renal and cardiovascular disease) in ~560,000 people (50% larger than the largest study to date; Haas et al. AJHG 2018), including ~50,000 with diabetes. A comprehensive array of follow-up analyses included electronic health record associations in an independent dataset, fine-mapping, multi-omics integration, and in vitro work. They identify a total of 68 associated loci (more than double the number in the largest analysis to date) and a number of candidate causal genes for these associations.

This is a high-quality study and a well-written paper. A few comments and suggestions for further improvement are below.

Response: Thank you for the positive feedback.

1-Inclusion/exclusion criteria for participants: From the Methods section, it is not immediately clear if people with clinically-defined microalbuminuria (MA) or those with diabetes mellitus (DM) have been included in the main meta-analysis or whether there were other inclusion/exclusion criteria. This should be noted at the beginning of the methods section in the part that describes the cohorts.

Response: We have now added this information to our manuscript on page 24, line 879 and on page 25, line 904.

2-Directionality of associations: It would be helpful if the authors could provide some additional analyses and insights on whether the genetic associations they observe are cause or consequence of major diseases outcomes related to kidney dysfunction (CKD, coronary disease, stroke, hypertension, heart failure), i.e. whether directionality of association is (a) gene variant -> impaired kidney function -> disease or (b) gene variant -> disease (e.g. DM) -> impaired kidney function. This could be achieved with a combination of: (1) Mendelian randomization analyses of polygenic score for UACR -> outcomes (using UK Biobank or summary statistics from previous GWAS); (2) Mendelian randomization analyses of disease polygenic scores to UACR; (3) analyses of effect size on UACR (in

people without disease, which could be done in UKBB at least) vs effect size on outcome (from UKBB or GWAS summary statistics); (4) identification of variants that appear to have a primary effect on disease risk and secondary on UACR vs those that appear to have a primary effect on UACR. This may also help interpret some of the potential for therapeutic translation of the GWAS findings.

Response: In order to complement the Mendelian randomization analyses published in Haas *et al.*¹ using the increased number of UACR loci identified in our study, we constructed a weighted genetic risk score based on the 59 index SNPs of our trans-ethnic meta-analysis of UACR for analyses with cardiovascular outcomes and diabetes as suggested by the Reviewer. In detail, we performed two-sample Mendelian Randomization using a weighted genetic risk score of the UACR-associated SNPs from our study and associating it with coronary artery disease from the published data of CARDIoGRAMplusC4D Consortium¹⁰, with stroke from the MEGASTROKE Consortium¹¹, and hypertension and heart failure assessed in the UKBB according to the definitions used in Haas *et al.* For the reverse analysis, we assessed an association of the two major CKD risk factors hypertension and type 2 diabetes (T2DM) with UACR. The corresponding genetic risk scores were constructed based on published index SNPs from a recent study analyzing blood pressure in over 1 million individuals¹² and from a meta-analysis of T2DM¹³.

A summary of these analyses is given in **Reviewer Table 1**, below. Our analyses support higher genetically determined UACR as related to hypertension after correcting for multiple testing. Conversely, both genetically higher systolic as well as diastolic blood pressure as well as T2DM are related to higher UACR. The bi-directional causal relationship between blood pressure/hypertension and higher UACR is consistent with the findings reported in Haas *et al*, and a feed-forward loop was provided as an explanation. The association of a T2DM genic risk score with UACR was highly significant ($p=1e-10$).

We have added these new results to the manuscript on page 14, line 546 (Results), and on page 27 line 967ff (Methods). We also included all results from genetic risk score analyses as a new **Supplementary Table 11**.

Reviewer Table 1: Association results of the genetic risk score analyses of UACR and its risk factors

Exposure	Outcome	N SNPs	Effect	SE	P-value	N cases	N total
UACR	CAD	57	0.003	0.085	0.975	60,801	184,305
UACR	HF	59	0.398	0.158	0.012	6,065	408,313
UACR	HTN	59	0.386	0.041	2.38E-21	226,011	408,539
UACR	Stroke	57	0.145	0.074	0.049	67,162	521,612
DBP	UACR	319	0.007	0.001	1.23E-24	-	757,601
SBP	UACR	244	0.008	0.001	3.52E-63	-	757,601
T2DM	UACR	138	0.018	0.003	1.01E-10	62,892	659,316

N cases: number of cases with having the disease; N total: total sample size of the non-UACR trait

CKD was not included in the risk score analyses, because UACR and CKD cannot be clearly separated. Not only is the UACR one parameter to define clinical CKD, but it also contains urinary creatinine in its denominator, which is strongly influenced by serum creatinine, used to define CKD in published GWAS. Thus, any risk score association analyses would be difficult to interpret and potentially misleading.

We acknowledge the additional points (3 and 4) brought up by the Reviewer. However, we strongly feel that addressing these research questions as well as including a comprehensive Mendelian Randomization analysis will be out of scope of our current work's aim and deserves a separate project that also thoroughly assesses how to best restrict the UKBB dataset to healthy individuals including information based on blood biomarkers.

3-Genetic association across the spectrum of UACR variation. It'd be interesting to know if the identified variants are associated with UACR homogeneously across the phenotypic variation of UACR in the population or if effect sizes tend to be stronger (or weaker) in people with extreme UACR. This can be done using conditional quantile regression and meta-regression as done by Abadi et al. (Am J Hum Genet. 2017;101:925-938) using individual level genotype data and UACR data from UK Biobank. Another complementary approach could be to divide the UKBB population in quantiles of the polygenic score and study the association with clinically-defined microalbuminuria for different quantiles relative to the bottom one (low-genetic risk). Both analyses could be quite insightful.

Response: To address this comment we chose the Reviewer's second suggestion for dividing the UKBB population in quartiles of the genic risk score and study the association with microalbuminuria status. As only <500 individuals were available with a diagnosis of microalbuminuria based on ICD-10 codes in the UKBB, we used the microalbuminuria definition that was also employed in our GWAS meta-analysis of MA, which is also the threshold used to define microalbuminuria clinically as well as to stage CKD. As indicated in **Reviewer Table 2**, the point estimates of the association of the UACR score quartiles with MA suggest a linear trend, whereas the confidence intervals overlap. The association results were added to the manuscript on page 13, line 504, and the result table was added as new **Supplementary Table 9**.

Reviewer Table 2: Associations between quartiles of a genetic risk score for UACR with MA in the UKBB.

Quartile	Effect	SE	Odds ratio (95% CI)	P-value	N cases	N controls
I			reference		6,281	54,096
II	0.159	0.019	1.17 (1.13-1.22)	2.24E-17	7,001	51,618
III	0.282	0.018	1.33 (1.28-1.38)	7.45E-53	7,583	49,799
IV	0.527	0.018	1.69 (1.64-1.75)	3.04E-191	9,149	47,224

We feel that further in-depth analyses of the heterogeneity of the UACR associated loci as well as their potential gene-environment effects according to Abadi *et al.* should be best studied in a dedicated project that includes a substantial proportion of individuals with macroalbuminuria as well.

Minor

-It has been disputed (see for instance Nat Genet. 2016;48:314-7) whether the conventional threshold for genome-wide significant 5×10^{-8} holds with more densely imputed panels (as in this study). Wonder if authors could report in a supplementary table how many loci would survive one of the more stringent thresholds that have been proposed or whether any additional variants would meet an annotation-specific threshold (lower for protein truncating variants or missense variants as suggested in the Nat Genet. 2016;48:314-7 paper).

Response: We agree that explicitly stating alternative significance thresholds can be useful for readers. Therefore, we have included alternative genome-wide significance thresholds with the corresponding references into the footnote of the main results table, Supplementary Table 3. These thresholds, 1×10^{-8} and 1.6×10^{-8} , are not met by 10 and 8 SNPs, respectively. All relevant tables are provided in excel format, to make it easy for readers to apply filters of their choice to the results.

-It'd be interesting to know how much of the (low) heritability of UACR is explained by the hits reported here. Please, calculate and report heritability explained.

Response: We have computed the heritability of UACR using the trans-ethnic summary statistics as 0.69% and have added this information to the manuscript (page 10, line 394 for the Results, and page 26, line 936 for the Methods).

-Fine mapping: the preliminary step of the fine-mapping uses approximate conditional analyses with GCTA, which is a reasonable first-line approach but is prone to error particularly for lower frequency variants where the pattern of LD is less precisely measured by reference panels and may be imperfectly captured by the r-squared. Some of the conditionally-independent variants identified by GCTA at the CUBN locus are about 1% allele frequency. At this locus and at the other ones where more than one signal are suggested by GCTA, the results of approximate conditional analyses should ideally be confirmed with formal conditional analyses (eg. using individual-level genotypes in UKBB). At the bare minimum, authors could show that the conditionally-independent variants put forward by GCTA are actually conditionally-independent of each other using individual-level data (a more relaxed p-value threshold can be used to account for the loss of sample size).

Response: As suggested by the Reviewer, we have now carried out conditional analyses based on individual-level data of 396,865 individuals in the UKBB by calculating both the single SNP linear regression, and a combined association model including all SNPs determined as independent based on our summary-statistics based conditional analyses in each of the three regions with multiple signals. As shown in **Reviewer Table 3**, only one SNP in the CUBN locus (rs141493439, MAF=1%) and another SNP in the FOXD2 locus (rs17453832, MAF=54%) showed a substantially attenuated effect size (>25% change) and a p-value>0.005 in the conditional analysis whereas the effects of some other SNPs, including low-frequency variants, in the locus became larger in this sensitivity analysis based on individual-level data. Since we cannot exclude that the changes in effect size might at least partially be related to the smaller sample size of the UKBB compared to our GWAS meta-analysis and/or to differences in LD across the studies, we decided not to exclude these two SNPs from our subsequent analyses, also since the use of GCTA in conjunction with a large LD reference panel to perform summary-statistics based conditional analyses is a commonly accepted method for accessing secondary signals. To inform readers of the results from this sensitivity analysis, we marked the two SNPs with attenuated effect estimates and added a corresponding footnote in Supplementary Table 13.

Reviewer Table 3: Sensitivity analyses of conditionally independent SNPs from summary statistics based on individual-participant data in a large contributing study.

SNPs in association model	reported SNP	locus	beta UKBB	SE UKBB	p UKBB	GCTA joined beta	GCTA joined p	% change beta UKBB
rs141493439	rs141493439	CUBN	0.215	0.011	4.60E-84	0.097	2.24E-10	
rs557338857+rs141493439+rs45551835+rs74375025+rs562661763	rs141493439	CUBN	0.047	0.023	4.41E-02	0.097	2.24E-10	-52%
rs45551835	rs45551835	CUBN	0.195	0.009	2.12E-95	0.106	3.98E-15	
rs557338857+rs141493439+rs45551835+rs74375025+rs562661763	rs45551835	CUBN	0.131	0.020	4.17E-11	0.106	3.98E-15	24%
rs557338857	rs557338857	CUBN	-0.138	0.016	2.50E-18	-0.104	4.26E-12	
rs557338857+rs141493439+rs45551835+rs74375025+rs562661763	rs557338857	CUBN	-0.114	0.016	1.93E-12	-0.104	4.26E-12	10%
rs562661763	rs562661763	CUBN	-0.134	0.014	2.39E-21	-0.092	1.80E-11	
rs557338857+rs141493439+rs45551835+rs562661763	rs562661763	CUBN	-0.107	0.015	2.19E-13	-0.092	1.80E-11	15%

rs74375025+rs562661763								
rs74375025	rs74375025	CUBN	0.059	0.004	2.07E-58	0.038	4.23E-28	
rs557338857+rs141493439+rs45551835+rs74375025+rs562661763	rs74375025	CUBN	0.037	0.004	1.22E-21	0.038	4.23E-28	-2%
rs1337526								
rs1337526	rs1337526	FOXD2	-0.026	0.003	4.90E-20	-0.024	1.87E-21	
rs17453832+rs1337526	rs1337526	FOXD2	-0.021	0.003	1.58E-09	-0.024	1.87E-21	-15%
rs17453832	rs17453832	FOXD2	-0.020	0.003	1.14E-13	-0.016	5.66E-10	
rs17453832+rs1337526	rs17453832	FOXD2	-0.009	0.003	6.07E-03	-0.016	5.66E-10	-45%
rs143200968								
rs143200968	rs143200968	HNRNPUL1	-0.042	0.008	4.26E-08	-0.039	2.12E-08	
rs143200968+rs15052	rs143200968	HNRNPUL1	-0.043	0.008	3.46E-08	-0.039	2.12E-08	8%
rs15052	rs15052	HNRNPUL1	0.015	0.003	1.92E-07	0.017	1.63E-10	
rs143200968+rs15052	rs15052	HNRNPUL1	0.015	0.003	1.56E-07	0.017	1.63E-10	-12%

-The abstract and discussion report 68 loci. After reading the main text, it seems that these stem from 59 trans-ethnic meta-analysis loci + 5 additional from ethnic specific analysis + 4 additional from diabetes-specific analysis. It would be helpful if this was explicitly stated in the main text at the beginning of the results and depicted in Figure S1.

Response: We have now included this information on page 10, line 372, and changed Figure S1 accordingly.

-In Figure S1, please add a panel C with the workflow of the diabetes-specific analysis. A separate additional panel D could be added with a Venn diagram showing the overlap of the identified loci in the three strands of the study.

Response: We added the requested panels to Figure S1.

-Line 441: for binomial test, please provide number of variants that were found to have a concordant direction.

Response: We have now clarified that all 59/59 SNPs showed concordant effect directions on page 11, line 427.

-Line 412: $p < 0.05$ and same direction? If some are in different direction, please report only the number of variants with directional concordance.

Response: Yes, since all 59/59 SNPs were direction consistent, none were in a different direction.

-Line 416: "mostly population based" should read "other cohorts, most of which are population-based". Correct?

Response: Yes, this is correct. We have changed the sentence accordingly.

-LDSC-regression intercept values should be used throughout in lieu of lambda GC, which does not distinguish genetic confounding from signal. Given the size of the study, attenuation ratio and mean chi-squared statistics should be reported in addition to LDSC-regression intercept values (as recommended in Nat Genet. 2018 Jul;50(7):906-908).

Response: As the LDSC-regression intercept was < 1 (0.95 after correction of a small rounding error in the LDSC-regression intercept value in the manuscript), the attenuation ratio would be < 0 (i.e. -0.12). The study in Nat Genet. 2018 Jul;50(7):906-908 reported the attenuation ratio to address the case of an LDSC-regression intercept rising above 1, and we have therefore not added this information to the manuscript.

-Line 424: “suggestive” should read “association that did not meet the genome-wide significance threshold”.

Response: We have rephrased this as suggested.

-Line 425: please provide p for each SNP. Could you test recessive inheritance or association in heterozygous or homozygous variant carriers versus homozygous non-carriers in UKBB?

Response: We have modified the section on *APOL1* as detailed in the response to Reviewer 1, comment 4.

-Line 494: “lipid” is misspelled.

Response: Please compare our response to minor comment 1 by Reviewer 1, who has raised a similar issue.

References

1. Haas, M. E. *et al.* Genetic Association of Albuminuria with Cardiometabolic Disease and Blood Pressure. *Am. J. Hum. Genet.* (2018). doi:10.1016/j.ajhg.2018.08.004
2. Thorner, P. S., Zheng, K., Kalluri, R., Jacobs, R. & Hudson, B. G. Coordinate gene expression of the alpha3, alpha4, and alpha5 chains of collagen type IV. Evidence from a canine model of X-linked nephritis with a COL4A5 gene mutation. *J. Biol. Chem.* **271**, 13821–8 (1996).
3. Hermle, T. *et al.* GAPVD1 and ANKFY1 Mutations Implicate RAB5 Regulation in Nephrotic Syndrome. *J. Am. Soc. Nephrol.* **29**, 2123–2138 (2018).
4. Wu, H. *et al.* Comparative Analysis and Refinement of Human PSC-Derived Kidney Organoid Differentiation with Single-Cell Transcriptomics. *Cell Stem Cell* **23**, 869–881.e8 (2018).
5. Park, J. *et al.* Single-cell transcriptomics of the mouse kidney reveals potential cellular targets of kidney disease. *Science* **360**, 758–763 (2018).
6. Ahlqvist, E., van Zuydam, N. R., Groop, L. C. & McCarthy, M. I. The genetics of diabetic complications. *Nat. Rev. Nephrol.* **11**, 277–87 (2015).
7. Zhang, Y., Qi, G., Park, J.-H. & Chatterjee, N. Estimation of complex effect-size distributions using summary-level statistics from genome-wide association studies across 32 complex traits. *Nat. Genet.* **50**, 1318–1326 (2018).
8. Wodarz, A., Ramrath, A., Grimm, A. & Knust, E. Drosophila atypical protein kinase C associates with Bazooka and controls polarity of epithelia and neuroblasts. *J. Cell Biol.* **150**, 1361–74 (2000).
9. Gamblin, C. L., Hardy, É. J.-L., Chartier, F. J.-M., Bisson, N. & Laprise, P. A bidirectional antagonism between aPKC and Yurt regulates epithelial cell polarity. *J. Cell Biol.* **204**, 487–95 (2014).
10. Nikpay, M. *et al.* A comprehensive 1,000 Genomes-based genome-wide association meta-analysis of coronary artery disease. *Nat. Genet.* **47**, 1121–1130 (2015).
11. Malik, R. *et al.* Multiancestry genome-wide association study of 520,000 subjects identifies 32 loci associated with stroke and stroke subtypes. *Nat. Genet.* **50**, 524–537 (2018).
12. Evangelou, E. *et al.* Genetic analysis of over 1 million people identifies 535 new loci associated with blood pressure traits. *Nat. Genet.* **50**, 1412–1425 (2018).

13. Xue, A. *et al.* Genome-wide association analyses identify 143 risk variants and putative regulatory mechanisms for type 2 diabetes. *Nat. Commun.* **9**, 2941 (2018).

Reviewers' Comments:

Reviewer #2:

Remarks to the Author:

Most of my comments are well-answered. The *Drosophila* experiments are now much better described. Personally, I am not happy with the idea that UAS-RNAi lines can be used without quantitative data (such as qPCR) to quantify the knockdown. However, observation of a particular phenotype with two independent lines for a single gene provides some reassurance. I'll leave that to the editors to decide.

Reviewer #3:

Remarks to the Author:

-The suggestion that fine-mapping should be restricted to European Ancestry participants "because a sufficiently large population to estimate reference LD was only available for this ancestry" seems strange. Surely, LD for African Americans, East or South Asians or Hispanics is well captured in the 1000 Genomes project or other existing reference panels. Please amend this sentence. An advantage of using multi-ancestry fine-mapping is that it can help narrow down credible sets due to different LD. At least for loci where in the non-EA ancestry subsets there was enough association signal, it would be important if fine-mapping was conducted in each set separately and perhaps combined. After all, it would give more prominence to the trans-ethnic component of the study.

-Thank you for adding Supplementary Table 11. However, from the table it is not clear what the "Effect" column represents. Is this a log odds ratio or a beta coefficient? For which unit of intermediate trait is this "effect" reported, per allele, per SD? Please provide both units as otherwise it's very difficult to understand what the table reports. For binary outcomes the OR and 95% CI should be reported.

-page 328: "word-wide" is a typo.

Dear Reviewers,

We thank you again for checking our revised manuscript and evaluating our responses to your comments. Below, please find a response to your most recent comments.

Sincerely,

Alexander Teumer, Cristian Pattaro, and Anna Köttgen, on behalf of all Authors

Point-by-point response

Reviewer #3:

-The suggestion that fine-mapping should be restricted to European Ancestry participants “because a sufficiently large population to estimate reference LD was only available for this ancestry” seems strange. Surely, LD for African Americans, East or South Asians or Hispanics is well captured in the 1000 Genomes project or other existing reference panels. Please amend this sentence. An advantage of using multi-ancestry fine-mapping is that it can help narrow down credible sets due to different LD. At least for loci where in the non-EA ancestry subsets there was enough association signal, it would be important if fine-mapping was conducted in each set separately and perhaps combined. After all, it would give more prominence to the trans-ethnic component of the study.

Response: Although we agree that using multi-ancestry fine-mapping can help narrow down credible sets in general, we need to point out that the implemented fine-mapping approach is not suitable for the suggested fine-mapping analyses. The implemented GCTA summary statistics-based method for fine-mapping requires at a minimum 2,000 individuals for estimating the reference LD. The sample size included in the 1000 Genomes reference panel will therefore be too small for obtaining reliable results, as the 1000 Genomes panel includes less than 700 individuals per super population. In addition, it is unlikely that these analyses will add much to the paper as the majority (>97%) of our GWAS populations are of European ancestry (also reflected by the revised title of the manuscript). As suggested by the Reviewer, we therefore modified the sentence in the manuscript, which now reads “These analyses were limited to EA, comprising >97% of the total sample, for whom large datasets to estimate reference LD for summary statistics-based fine-mapping are publicly accessible” (page 14, lines 780-781).

-Thank you for adding Supplementary Table 11. However, from the table it is not clear what the “Effect” column represents. Is this a log odds ratio or a beta coefficient? For which unit of intermediate trait is this “effect” reported, per allele, per SD? Please provide both units as otherwise it’s very difficult to understand what the table reports. For binary outcomes the OR and 95% CI should be reported.

Response: Upon the Reviewer’s request, we added a column containing the odds ratio and 95% CI in case of a binary outcome, and added a column describing the interpretation of the reported effect size (now Supplementary Table 2). The effect of the genetic risk score represents the change of the outcome per unit change of the (genetically estimated) exposure.

-page 328: “word-wide” is a typo.

Response: Thank you for spotting this. We corrected the typo.